# Constrained Meta Reinforcement Learning with Provable Test-Time Safety

**Tingting Ni** [1]  **Maryam Kamgarpour** [1]

## Abstract

Meta reinforcement learning (RL) allows agents to leverage experience across a distribution of tasks on which the agent can train at will, enabling faster learning of optimal policies on new test tasks. Despite its success in improving sample complexity on test tasks, many real-world applications, such as robotics and healthcare, impose safety constraints during testing. Constrained meta RL provides a promising framework for integrating safety into meta RL. An open question in constrained meta RL is how to ensure safety of the policy on the real-world test task, while reducing the sample complexity and thus, enabling faster learning of optimal policies. To address this gap, we propose an algorithm that refines policies learned during training, with provable safety and sample complexity guarantees for learning a near optimal policy on the test tasks. We further derive a matching lower bound, showing that this sample complexity is tight.

## 1. Introduction

Reinforcement learning (RL) has achieved remarkable success in domains such as games (Mnih et al., 2013; Silver et al., 2016), language model fine-tuning (Ouyang et al., 2022), and robotics (Song et al., 2021). Despite these successes, several challenges remain. Two of the main challenges are: 1) sample complexity, that is, reducing the number of interactions with the real-world system to learn an optimal policy; and 2) safety, that is, ensuring constraint satisfaction such as collision avoidance in robotics (Koppejan & Whiteson, 2011) and operational restrictions in healthcare (Kyrarini et al., 2021).

A promising approach to address the first challenge is meta RL (Ghavamzadeh et al., 2015), where the agent learns across a distribution of tasks, each modeled as a Markov decision process. Learning proceeds in two stages: a training phase in simulation and a test phase in the real world. Simulation enables inexpensive data collection, as in robotics (Makoviychuk et al., 2021), autonomous driving (Kazemkhani et al., 2025), and healthcare (Visentin et al., 2014). The knowledge acquired during training is then leveraged to accelerate learning on a new test task drawn from the same distribution.

Meta RL has demonstrated empirical success in reducing the number of real-world interactions required to optimize policies compared to standard RL methods (Finn et al., 2017; Rakelly et al., 2019; Beck et al., 2025). More recently, several works have provided theoretical guarantees for these improvements (Ye et al., 2023; Chen et al., 2022; Mutti & Tamar, 2024). In particular, standard RL without a training phase requires at least $\tilde{\mathcal{O}}(\varepsilon^{-2}|\mathcal{S}||\mathcal{A}|)$ samples to learn an $\varepsilon$-optimal policy (He et al., 2021), where $|\mathcal{S}|$ and $|\mathcal{A}|$ denote the cardinality of the state and action spaces. In contrast, Ye et al. show that learning a near-optimal policy requires only $\tilde{\mathcal{O}}(\varepsilon^{-2}\mathcal{C}(\mathcal{D}))$ samples from the test task, where $\mathcal{C}(\mathcal{D})$ quantifies the complexity of the task distribution $\mathcal{D}$ over all tasks. In particular, $\mathcal{C}(\mathcal{D})$ can be significantly smaller than the sizes of the state and action spaces when the distribution is concentrated (e.g., sub-Gaussian) or when the set of tasks is small (Ye et al., 2023). Subsequent works further improve this sample complexity under additional structural assumptions on the set of tasks, such as separation (Chen et al., 2022) or strong identifiability (Mutti & Tamar, 2024).

To address the safety challenge, constrained RL incorporates safety constraints directly into the learning process by modeling problems as constrained Markov decision processes. A central requirement is safe exploration, which ensures that constraints are satisfied throughout testing, particularly when violations are costly or dangerous, such as in real-world safety-critical systems (Koller et al., 2018).

Safe exploration in constrained RL has been widely studied using both model-based (Berkenkamp et al., 2021; Chow et al., 2018; Wachi et al., 2018) and model-free approaches (Achiam et al., 2017; Milosevic et al., 2025). Among works providing high-probability guarantees for learning an $\varepsilon$-optimal policy under safe exploration, model-based methods (Yu et al., 2025; Bura et al., 2022; Liu et al., 2021) iteratively solve linear programs based on the learned dy-

[1]Sycamore Lab, EPFL, Lausanne, Switzerland. Correspondence to: Tingting Ni <tingting.ni@epfl.ch>.

*Proceedings of the $43^{rd}$ International Conference on Machine Learning*, Seoul, South Korea. PMLR 306, 2026. Copyright 2026 by the author(s).

namics, resulting in a sample complexity that scales polynomially with the size of the state and action spaces, namely $\tilde{\mathcal{O}}(\varepsilon^{-2}\text{poly}(|\mathcal{S}|,|\mathcal{A}|))$. In contrast, Ni & Kamgarpour propose a model-free approach that directly estimates policy gradients from interactions, avoiding the estimation of the dynamics but incurring a worse dependence on $\varepsilon$, with sample complexity $\tilde{\mathcal{O}}(\varepsilon^{-6})$. However, reducing real-world interactions is even more critical in constrained settings, where interactions may lead to violations of safety constraints. This motivates constrained meta RL.

Constrained meta RL extends constrained RL by introducing a training phase over a task distribution to accelerate learning on a new test task. During training, a so-called meta policy is learned to maximize the expected reward while satisfying safety constraints either in expectation (Cho & Sun, 2024; Khattar et al., 2023) or uniformly across tasks (Xu & Zhu, 2026; As et al., 2026). Some of these works further study how to adapt the meta policy from training to testing so that it becomes optimal for the test task. Specifically, Cho & Sun employ constrained policy optimization and guarantee that the adapted policy satisfies the constraints for the test task; however, safe exploration during the test phase is not addressed. Later, Xu & Zhu develop a closed-form method for policy adaptation, and establish guarantees for safe exploration and monotonic reward improvement under the assumption that the test task is known.

Inspired by the theoretical success of meta RL in provably improving sample complexity on test tasks, we study constrained meta RL and leverage training knowledge to initialize a safe policy and iteratively refine it toward optimality, ensuring test-time safety with minimal sample complexity.

1. Our algorithm builds upon (Ye et al., 2023), which considers the unconstrained meta RL setting. Our algorithm learns an $\mathcal{O}(\varepsilon)$-optimal policy while ensuring safe exploration with high probability, using $\tilde{\mathcal{O}}(\varepsilon^{-2}\mathcal{C}(\mathcal{D}))$ test task samples (Corollary 4.3). In Appendix A, Table 1 summarizes different sample complexity guarantees.

2. By constructing hard instances, we establish a matching problem-dependent lower bound on the sample complexity required to obtain an $\varepsilon$-optimal and feasible policy for the test task (Theorem 5.1).

3. We empirically validate our approach in a gridworld environment, where test tasks are unknown and sampled from a truncated Gaussian distribution. Our method outperforms both constrained RL baselines and a constrained meta RL baseline in terms of learning efficiency, while ensuring safe exploration.

## 2. Problem Formulation

We first present the necessary background on constrained Markov decision processes (CMDPs) and constrained meta RL, before formally introducing the learning problem addressed in this paper.

**Notation:** Let $\mathbb{N}$ and $\mathbb{R}$ denote the sets of natural numbers and real numbers, respectively. For a set $\mathcal{X}$, $\Delta(\mathcal{X})$ denotes the probability measure over $\mathcal{X}$, and $|\mathcal{X}|$ denotes its cardinality. For any $p, q \in \Delta(\mathcal{X})$, their total variation distance is defined as $\|p - q\|_{\text{TV}} := \frac{1}{2}\sum_{x\in\mathcal{X}}|p(x) - q(x)|$. For any integer $m$, we set $[m] := \{1,\ldots,m\}$. We use $\|\cdot\|_\infty$ to denote the supremum norm. For any scalar $z \in \mathbb{R}$, we denote its positive part by $[z]_+ := \max\{z, 0\}$, and its negative part by $[z]_- := \max\{-z, 0\}$. For an event $\mathcal{E}$, let $\mathbf{1}[\mathcal{E}]$ denote the indicator function, which equals 1 if and only if $\mathcal{E}$ is true.

### 2.1. Constrained Markov Decision Processes

An infinite-horizon CMDP[1] is defined by the tuple:

$$\mathcal{M}_i = \{\mathcal{S}, \mathcal{A}, \rho_i, P_i, r_i, c_i, \gamma\}.$$

We refer to $\mathcal{M}_i$ as a task, where $i$ is the task index. The sets $\mathcal{S}$ and $\mathcal{A}$ denote the (possibly continuous) state and action spaces, $\rho_i \in \Delta(\mathcal{S})$ is the initial state distribution, and $P_i(s' \mid s, a)$ is the transition probability from state $s$ to $s'$ given action $a$. The reward function is $r_i : \mathcal{S} \times \mathcal{A} \to [0, 1]$, and the constraint function is $c_i : \mathcal{S}\times\mathcal{A} \to [-1, 1]$. Both the reward and constraint functions are assumed to be bounded. Here, $\gamma \in (0, 1)$ is the discount factor.

A stationary Markovian policy $\pi : \mathcal{S} \to \Delta(\mathcal{A})$ maps each state to a probability distribution over the action space. Let $\Pi$ denote the set of all such policies. Starting from $s_0 \sim \rho_i$, at each timestep $t$, the agent takes an action $a_t \sim \pi(\cdot \mid s_t)$, transitions to $s_{t+1} \sim P_i(\cdot \mid s_t, a_t)$, and receives reward $r_i(s_t, a_t)$ and constraint $c_i(s_t, a_t)$. The resulting trajectory is $\tau = (s_0, a_0, s_1, a_1, \ldots) \in (\mathcal{S} \times \mathcal{A})^\infty$. For any policy $\pi$ and task $\mathcal{M}_i$, we define the cumulative discounted reward as $V_r^{\mathcal{M}_i}(\pi) := \mathbb{E}_{\tau\sim\pi}\left[\sum_{t=0}^\infty \gamma^t r_i(s_t, a_t)\right]$, and the constraint value as $V_c^{\mathcal{M}_i}(\pi) := \mathbb{E}_{\tau\sim\pi}\left[\sum_{t=0}^\infty \gamma^t c_i(s_t, a_t)\right]$.

Given a CMDP $\mathcal{M}_i$, the agent's objective is to find a policy that maximizes the expected reward subject to the constraint:

$$\max_{\pi\in\Pi} V_r^{\mathcal{M}_i}(\pi) \quad \text{s.t.} \quad V_c^{\mathcal{M}_i}(\pi) \geq 0. \qquad \text{(CMDP}_i\text{)}$$

We assume the existence of an optimal Markovian policy. In finite action spaces, this is guaranteed under Slater's condition (Altman, 2021), stated as follows:

**Assumption 2.1** (Slater's condition). Given $\mathcal{M}_i$, there exists a positive constant $\xi_i$ and a policy $\pi \in \Pi$ such that $V_c^{\mathcal{M}_i}(\pi) \geq \xi_i$.

---

[1]Our results directly extend to the finite-horizon setting. Here, we focus on the infinite-horizon discounted setting as it is more commonly considered in deep RL.

In continuous action spaces $\mathcal{A}$, the existence of an optimal Markovian policy is guaranteed under additional measurability and compactness assumptions on $\mathcal{A}$ (Hernández-Lerma & González-Hernández, 2000, Theorem 3.2).

Next, we introduce two concepts, which will be used for designing the algorithm in Section 3.1.

**Feasibility of policies for $\mathcal{M}_i$:** We classify the policies based on the extent to which they satisfy the constraints. Given $\xi \geq 0$, a policy $\pi$ is *relaxed $\xi$-feasible* if $V_c^{\mathcal{M}_i}(\pi) \geq -\xi$, *$\xi$-feasible* if $V_c^{\mathcal{M}_i}(\pi) \geq \xi$, and *feasible* if $\xi$ is zero.

**Distance between CMDPs:** For any two CMDPs $\mathcal{M}_i$ and $\mathcal{M}_j$, Feng et al. define the distance $d(\mathcal{M}_i, \mathcal{M}_j)$ as

$$d(\mathcal{M}_i, \mathcal{M}_j) := \max \Big\{ \|r_i - r_j\|_\infty, \ \|c_i - c_j\|_\infty,$$

$$\|\rho_i - \rho_j\|_{\mathrm{TV}}, \max_{(s,a) \in \mathcal{S} \times \mathcal{A}} \|P_i(\cdot \mid s, a) - P_j(\cdot \mid s, a)\|_{\mathrm{TV}} \Big\}.$$

This metric captures the largest difference across all components of the two CMDPs, using the supremum norm for bounded functions ($r_i$ and $c_i$) and total variation distance for probability distributions ($\rho_i$ and $P_i$). We write $\mathcal{M}_i \in B(\mathcal{M}_j, \varepsilon)$ if $d(\mathcal{M}_i, \mathcal{M}_j) \leq \varepsilon$.

## 2.2. Constrained Meta Reinforcement Learning

In constrained meta RL, we are given a family of tasks denoted by $\mathcal{M}_{\mathrm{all}} := \{\mathcal{M}_i\}_{i \in \Omega}$, where $\Omega$ is a large and possibly uncountable index set. The main distinction from standard meta RL (Finn et al., 2017) is the presence of the constraint function $V_c^{\mathcal{M}_i}$ for each task $\mathcal{M}_i$. Such setting naturally arises in applications such as autonomous navigation and robotic systems with collision avoidance, where different goals and obstacles affect both rewards and constraints. Moreover, transition dynamics parameters, such as friction coefficients and joint damping, often vary over continuous ranges, resulting in an uncountable family of tasks.

Similar to meta RL, constrained meta RL proceeds in two phases: a *training phase* and a *testing phase*.

**The training phase** is performed offline (e.g., using simulators), where constraint violations are allowed and agents can freely sample CMDPs $\mathcal{M}_i$ with $i \sim \mathcal{D}$. With an abuse of notation, we also write $\mathcal{M}_i \sim \mathcal{D}$. For example, in dexterous hand manipulation, parameters such as surface friction or joint damping are often drawn from uniform, log-uniform, or truncated Gaussian distributions over bounded intervals (Andrychowicz et al., 2020). Given the availability of a simulator, we assume access to the following sampling model:

**Generative model:** Given a CMDP $\mathcal{M}_{\mathrm{train}}$, the oracle returns, for any $(s, a) \in \mathcal{S} \times \mathcal{A}$, the next state $s' \sim P(\cdot \mid s, a)$ together with the associated reward and constraint values $(r(s, a), c(s, a))$.

**The testing phase** occurs in the real world, where a new

and unknown CMDP $\mathcal{M}_{\mathrm{test}}$ is drawn from $\mathcal{D}$. In contrast to the training phase, we assume access only to a sampling model that generates trajectories under a given policy:

**online model:** Given a CMDP $\mathcal{M}_{\mathrm{test}}$ and a policy $\pi$, applying $\pi$ generates a trajectory of length $H$, $\tau_H := \big(s_t, a_t, r(s_t, a_t), c(s_t, a_t)\big)_{t=0}^{H-1}$, where $s_0 \sim \rho$, $a_t \sim \pi(\cdot \mid s_t)$, and $s_{t+1} \sim P(\cdot \mid s_t, a_t)$ for all $t \in \{0, \ldots, H-1\}$.

This online model is less stringent than the model used in training. Since constraint violations are not allowed in the real world, we require *safe exploration*, defined as follows:

**Definition 2.2** (Safe exploration). Given an algorithm that produces a sequence of policies $\{\pi_k\}_{k=1}^K$, we say it ensures safe exploration on $\mathcal{M}_{\mathrm{test}}$ if $\pi_k$ is feasible for all $k \in [K]$.

**Objective:** Our goal is to leverage knowledge acquired during training to gradually update the learned policy toward optimality for a new, unseen CMDP $\mathcal{M}_{\mathrm{test}}$ using minimal real-world interactions. This process is referred to as *policy adaptation* (Finn et al., 2017). To quantify the sample complexity of adaptation, where each sample corresponds to a single query with $\mathcal{M}_{\mathrm{test}}$ at each iteration, we consider the *test-time reward regret* (Efroni et al., 2020):

$$\mathrm{Reg}_r^{\mathcal{M}_{\mathrm{test}}}(K) := \sum_{k=1}^K [V_r^{\mathcal{M}_{\mathrm{test}}}(\pi_{\mathrm{test}}^*) - V_r^{\mathcal{M}_{\mathrm{test}}}(\pi_k)]_+. \quad (1)$$

This metric quantifies the cumulative sub-optimality of the deployed policies $\{\pi_k\}_{k=1}^K$. We now introduce the notion of a *mixture policy*, which forms a convex combination of the policies $\{\pi_k\}_{k=1}^K$ and will be central to our analysis of regret and feasibility.

**Definition 2.3.** A *mixture policy* is defined as $\pi := \sum_{k=1}^K d_k \pi_k$, where $\{\pi_k\}_{k=1}^K \subset \Pi$ and $\{d_k\}_{k=1}^K \in \Delta(K)$. It is implemented by sampling an index $i \sim \{d_k\}_{k=1}^K$ once at the beginning and then executing the corresponding policy $\pi_i$ for all timesteps.

Note that a mixture policy is Markovian, as its actions depend only on the current state and not on past trajectory information. Given this definition, sublinear reward regret implies that $\sum_{k=1}^K \pi_k / K$ converges to the optimal policy, providing a direct characterization of the sample complexity.[2]

## 3. Algorithm Design

In this section, we present algorithms for the training and testing phases that minimize test-time regret while ensuring safe exploration. Our approach is inspired by the Policy Collection–Elimination (PCE) algorithm of (Ye et al., 2023), originally developed for the unconstrained setting. In the

---

[2]In meta RL, Ye et al. show that knowing $\mathcal{D}$ provides only a constant improvement in test-time reward regret; nevertheless, we assume $\mathcal{D}$ is unknown for generality.

training phase, PCE learns a policy set such that, for any $\mathcal{M} \sim \mathcal{D}$, the optimal policy lies within a small neighborhood of this set. During testing, PCE sequentially executes policies from this set to identify a policy that is near-optimal for the test CMDP.

PCE provably reduces test-time reward regret by eliminating the need to learn an optimal policy from scratch. However, this approach is insufficient in the constrained setting, where safe exploration additionally requires every deployed policy to be feasible. Since the test environment $\mathcal{M} \sim \mathcal{D}$ is unknown a priori, we must ensure that the policy set contains at least one policy that is safe across all environments simultaneously. In general, an optimal policy for a given $\mathcal{M}$ need not be feasible for all $\mathcal{M} \sim \mathcal{D}$.

To address this, in Section 3.1, we modify the training phase of the PCE algorithm to learn not only a policy set that approximates the optimal policy for any $\mathcal{M} \sim \mathcal{D}$, but also a single policy that is feasible for all $\mathcal{M} \in \mathcal{M}_{\text{all}}$. In Section 3.2, we propose a novel algorithm that refines these policies via an adaptive mixture scheme, enabling efficient adaptation with safe exploration guarantees during testing.

### 3.1. Training Phase

To learn not only a policy set that approximates the optimal policy for any $\mathcal{M} \sim \mathcal{D}$ but also a single policy that is feasible for all $\mathcal{M} \in \mathcal{M}_{\text{all}}$, we first construct a CMDP set $\mathcal{U}$. If $\mathcal{U}$ is sufficiently large, in the sense that it covers the support of the task distribution with high probability, then any CMDP $\mathcal{M}_{\text{new}} \sim \mathcal{D}$ lies within an $\varepsilon$-neighborhood of some $\mathcal{M} \in \mathcal{U}$. For each $\mathcal{M} \in \mathcal{U}$, we learn a near-optimal policy, and in addition, we learn a single policy $\pi_s$ that is feasible for all CMDPs in $\mathcal{U}$.

The training phase relies on three oracles, which are based on recent advances in constrained RL (Achiam et al., 2017; Ding et al., 2020; Xu et al., 2021) and meta constrained RL (Xu & Zhu, 2026; As et al., 2026).

**Definition 3.1** (CMDP check oracle). Given two CMDPs $\mathcal{M}_1$ and $\mathcal{M}_2$, an accuracy parameter $\varepsilon > 0$, and a confidence level $\delta \in (0, 1)$, the oracle $\mathbb{O}_c(\mathcal{M}_1, \mathcal{M}_2, \varepsilon, \delta)$ returns 1 with probability at least $1 - \delta$ if $\mathcal{M}_1 \in B(\mathcal{M}_2, \varepsilon)$, and returns 0 otherwise.

For finite state and action spaces, $\mathbb{O}_c$ can be implemented by computing the distance $d(\mathcal{M}_1, \mathcal{M}_2)$ via enumeration, returning 1 if $d(\mathcal{M}_1, \mathcal{M}_2) \leq \varepsilon$. In large finite or continuous spaces, this distance can be estimated through sampling. Standard concentration arguments characterize the resulting estimation error as a function of the number of samples from $\mathcal{M}_1$ and $\mathcal{M}_2$, see (As et al., 2026, Theorem C.8).

Given a collection $\{\mathcal{M}_i\}_{i\in[N]}$, we apply $\mathbb{O}_c$ within Subroutine 3, adapted from (Ye et al., 2023, Algorithm 4), to construct a reduced set $\mathcal{U}$. Specifically, this subroutine starts from an empty set $\mathcal{U}$ and adds a CMDP $\mathcal{M}_{\text{new}}$ from $\{\mathcal{M}_i\}_{i\in[N]}$ only if $\mathbb{O}_c(\mathcal{M}_{\text{new}}, \mathcal{M}, \varepsilon/N^2, \delta) = 0$ for every $\mathcal{M} \in \mathcal{U}$, indicating that $\mathcal{M}_{\text{new}}$ is at least $\varepsilon$ away from all existing CMDPs in $\mathcal{U}$. Upon termination of Subroutine 3, $\mathcal{U}$ is a reduced set of $\{\mathcal{M}_i\}_{i\in[N]}$ in which any two CMDPs are separated by at least $\varepsilon$. For completeness, the corresponding pseudocode is provided in Appendix C.1.

**Definition 3.2** (Optimal policy oracle). Given CMDP $\mathcal{M}$, an accuracy parameter $\varepsilon > 0$, and a confidence level $\delta \in (0, 1)$, the oracle $\mathbb{O}_l(\mathcal{M}, \varepsilon, \delta)$ returns a policy $\pi$ along with its reward and constraint values $\{V_r^{\mathcal{M}}(\pi), V_c^{\mathcal{M}}(\pi)\}$ such that, with probability at least $1 - \delta$, $\pi$ is $\varepsilon$-optimal and relaxed $\varepsilon$-feasible for $\mathcal{M}$.

Above, we only require learning a near-optimal policy that is relaxed $\varepsilon$-feasible, rather than strictly feasible. This relaxation allows the use of a broader class of constrained RL algorithms (Achiam et al., 2017; Ding et al., 2020; Xu et al., 2021) compared with methods that require learning strictly feasible and optimal policies (Ni & Kamgarpour, 2025; Bura et al., 2022). Note that for $\mathcal{M}_1 \in B(\mathcal{M}_2, \varepsilon)$, the optimal policy for $\mathcal{M}_1$ corresponds to a near-optimal policy and relaxed $\mathcal{O}(\varepsilon)$-feasible for $\mathcal{M}_2$ (see Lemma C.4).

Next, given a set of CMDPs $\{\mathcal{M}_i\}_{i\in[N]} \subset \mathcal{M}_{\text{all}}$, our goal is to find a policy simultaneously feasible for all $\mathcal{M}_i$. To this end, we impose the following assumption, used in prior work on meta constrained RL (Xu & Zhu, 2026):

**Assumption 3.3** (Simultaneous Slater's condition). There exists a positive constant $\xi$ and a policy $\pi \in \Pi$ such that $\pi$ is $\xi$-feasible for every $\mathcal{M} \in \mathcal{M}_{\text{all}}$.

Assumption 3.3 guarantees the existence of a policy satisfying the constraints with a uniform positive margin $\xi$ across all CMDPs in $\mathcal{M}_{\text{all}}$. For example, in autonomous driving, it holds if there exists a policy that avoids all collisions in every task CMDP. Under this assumption, we can implement the simultaneously feasible policy oracle defined below:

**Definition 3.4** (Simultaneously feasible policy oracle). Given a finite set of CMDPs $\{\mathcal{M}_i\}_{i\in[N]}$, a safety margin $\xi > 0$, and a confidence level $\delta \in (0, 1)$, the oracle $\mathbb{O}_s(\{\mathcal{M}_i\}_{i\in[N]}, \xi, \delta)$ returns a policy $\pi$ along with its reward and constraint values $\{V_r^{\mathcal{M}_i}(\pi), V_c^{\mathcal{M}_i}(\pi)\}_{i\in[N]}$ such that, with probability at least $1 - \delta$, $\pi$ is $\xi$-feasible for all $\mathcal{M}_i$.

The oracle $\mathbb{O}_s$ can be implemented using existing algorithms (As et al., 2026; Xu & Zhu, 2026). For instance, Xu & Zhu construct a policy that is $\xi$-feasible for every $\{\mathcal{M}_i\}_{i\in[N]}$ via a policy gradient method. At each iteration, the gradient is computed with respect to the CMDP in which the current policy incurs the largest constraint violation. By focusing on the worst-case CMDP at each step, this method provides a tractable way to implement the oracle $\mathbb{O}_s$.

With these oracles in place, we introduce the training phase in Algorithm 1. The training phase iteratively samples

---

**Algorithm 1** Training phase

---

1: **Input:** Oracles $\mathbb{O}_l, \mathbb{O}_c, \mathbb{O}_s$; confidence level $\delta$; initial sample size $N = \ln^2(\delta)/\delta^2$; accuracy parameter $\varepsilon$; safety margin $\xi$.
2: **for** iteration $l = 1, 2, \ldots$ **do**
3: $\quad$ Sample $N$ CMDPs from $\mathcal{D}$ as $\{\mathcal{M}_i\}_{i \in [N]}$
4: $\quad$ Call $\mathbb{O}_c$ to construct $\mathcal{U}$ using Subroutine 3
5: $\quad$ **if** $\sqrt{|\mathcal{U}| \ln(2N/\delta)(N - |\mathcal{U}|)^{-1}} > \delta$ **then**
6: $\quad\quad$ $N = 2N$
7: $\quad$ **else**
8: $\quad\quad$ **for** $\forall \mathcal{M}_i \in \mathcal{U}$ **do**
9: $\quad\quad\quad$ Call $\mathbb{O}_l(\mathcal{M}_i, \varepsilon, \frac{\delta}{2|\mathcal{U}|}) \to \{\pi_i, V_r^{\mathcal{M}_i}(\pi_i), V_c^{\mathcal{M}_i}(\pi_i)\}$
10: $\quad\quad$ **end for**
11: $\quad\quad$ Call $\mathbb{O}_s(\mathcal{U}, \xi, \frac{\delta}{2}) \to \{\pi_s, \{V_r^{\mathcal{M}_i}(\pi_s), V_c^{\mathcal{M}_i}(\pi_s)\}_{\mathcal{M}_i \in \mathcal{U}}\}$
12: $\quad\quad$ Return $\pi_s$ and $\hat{\mathcal{U}}$ computed as per Eq. (2)
13: $\quad$ **end if**
14: **end for**

---

CMDPs from $\mathcal{D}$ to construct a CMDP set $\mathcal{U}$. In the first iteration (line 3), we sample $N := \ln^2(\delta)/\delta^2$ CMDPs to form $\{\mathcal{M}_i\}_{i \in [N]}$. In line 4, Subroutine 3 is applied to obtain a reduced set $\mathcal{U}$. In line 5, the algorithm evaluates the quantity $\sqrt{|\mathcal{U}| \ln(2N/\delta)/(N - |\mathcal{U}|)}$, which upper-bounds the error in estimating the coverage of $\mathcal{U}$ over the task distribution $\mathcal{D}$. If this error is greater than $\delta$, then $N$ is doubled and the procedure repeats. Otherwise, the set $\mathcal{U}$ constructed in line 4 is such that, with high probability, any CMDP from $\mathcal{D}$ lies within an $\varepsilon$-neighborhood of some CMDP in $\mathcal{U}$. Once $\mathcal{U}$ is constructed, we proceed as follows:

1. For each $\mathcal{M}_i \in \mathcal{U}$, call $\mathbb{O}_l(\mathcal{M}_i, \varepsilon, \delta/2|\mathcal{U}|)$ to obtain its near-optimal policy $\pi_i$ along with the associated reward and constraint values $\{V_r^{\mathcal{M}_i}(\pi_i), V_c^{\mathcal{M}_i}(\pi_i)\}$.

2. Call $\mathbb{O}_s(\mathcal{U}, \xi, \delta/2)$ to obtain a feasible policy $\pi_s$ and the associated reward and constraint values for all CMDPs in $\mathcal{U}$, i.e., $\{V_r^{\mathcal{M}_i}(\pi_s), V_c^{\mathcal{M}_i}(\pi_s)\}_{\mathcal{M}_i \in \mathcal{U}}$.

Given this information, we construct a policy-value set:

$$\hat{\mathcal{U}} := \left\{ (\pi, V_r^{\mathcal{M}}(\pi), V_c^{\mathcal{M}}(\pi), V_r^{\mathcal{M}}(\pi_s), V_c^{\mathcal{M}}(\pi_s)) \,\middle|\, \mathcal{M} \in \mathcal{U} \right\}, \quad (2)$$

Below, we characterize the near-optimality of $\hat{\mathcal{U}}$, assess the feasibility of $\pi_s$, and quantify the efficiency of Algorithm 1 in terms of iteration and CMDP samples. Our bounds depend on $C_\varepsilon(\mathcal{D}, \delta)$, a finite quantity measuring the complexity of the task distribution $\mathcal{D}$; its formal definition is given in Definition 4.1, with further discussion in Section 4.

**Lemma 3.5.** *Let Assumption 3.3 hold and set $(8L + 18)\varepsilon \leq \xi$, where $L := (1 - \gamma)^{-1} + 2\gamma(1 - \gamma)^{-2}$. With probability at least $1 - 25\delta - 9\delta \ln C_\varepsilon(\mathcal{D}, \delta)$, the following holds:*

*1. For any $\mathcal{M} \sim \mathcal{D}$, $\hat{\mathcal{U}}$ contains a policy-*

*value tuple $(\pi, u, v, u_s, v_s)$ such that $\pi$ is $4\varepsilon L(1 + \xi^{-1}(1 - \gamma)^{-1})$-optimal and*

$$\max\Big\{ |V_r^{\mathcal{M}}(\pi) - u|, |V_c^{\mathcal{M}}(\pi) - v|,$$
$$|V_r^{\mathcal{M}}(\pi_s) - u_s|, |V_c^{\mathcal{M}}(\pi_s) - v_s| \Big\} \leq \varepsilon L.$$

*2. For any $\mathcal{M} \sim \mathcal{D}$, the policy $\pi_s$ is $(\xi - \varepsilon L)$-feasible.*

*3. Algorithm 1 terminates with at most $\mathcal{O}(\ln C_\varepsilon(\mathcal{D}, \delta))$ iterations and samples at most $\tilde{\mathcal{O}}(\delta^{-2} C_\varepsilon(\mathcal{D}, \delta))$ CMDPs from $\mathcal{M}_{\text{all}}$, returning a policy-value set satisfying $|\hat{\mathcal{U}}| \leq 2C_\varepsilon(\mathcal{D}, \delta) \ln \delta^{-1}$.*

Lemma 3.5 extends (Ye et al., 2023, Lemma C.1) from the unconstrained to the constrained setting and from countable to uncountable CMDP sets. Its proof relies on two main techniques: (i) concentration bounds (Chernoff and union bounds) to ensure that, for any $\mathcal{M} \sim \mathcal{D}$, a sufficiently large CMDP set $\mathcal{U}$ has an $\varepsilon$-neighborhood containing $\mathcal{M}$ with high probability, and (ii) continuity results for the optimal values of constrained optimization problems, combined with the simulation lemma (Abbeel & Ng, 2005), to extend the near-optimality and feasibility of policies computed on $\hat{\mathcal{U}}$ to all CMDPs in $\mathcal{M}_{\text{all}}$. The proof is provided in Appendix C.2.

Observe that Algorithm 1, including all oracle calls, is executed in the training environment, where samples are relatively inexpensive due to the availability of a simulator. Combined with the third point of Lemma 3.5, this ensures that Algorithm 1 can be implemented in finite time. Furthermore, all oracles apply to both finite and continuous state and action spaces, as the underlying methods support such settings.

### 3.2. Testing Phase

From Lemma 3.5, for any test CMDP $\mathcal{M}_{\text{test}} \sim \mathcal{D}$, the policy–value set $\hat{\mathcal{U}}$ contains a policy that is near-optimal for $\mathcal{M}_{\text{test}}$. The goal of the testing phase is to efficiently identify such a policy while maintaining safe exploration. Directly executing a policy $\pi_l \in \hat{\mathcal{U}}$ may however violate the constraints in $\mathcal{M}_{\text{test}}$. On the other hand, the feasible policy $\pi_s$ guarantees safety but is generally suboptimal. To balance safety and reward, we introduce a *mixture policy*:

$$\pi_{l,m} = (1 - \alpha_{l,m})\pi_s + \alpha_{l,m}\pi_l,$$

where $\alpha_{l,m} \in [0, 1]$, indexed by the update counter $m$, is gradually increased to shift probability mass toward the candidate policy $\pi_l$ while preserving feasibility. The main challenges are (i) selecting an appropriate policy $\pi_l$ and (ii) adaptively updating $\alpha_{l,m}$ based on interactions with $\mathcal{M}_{\text{test}}$. We address these challenges in Algorithm 2, which proceeds in three stages: optimistic selection of a candidate policy $\pi_l$, verification of $\pi_l$, and adaptive updates of $\alpha_{l,m}$.

**Algorithm 2** Testing phase with safe exploration

1: **Input:** Policy–value set $\hat{\mathcal{U}}$ and $\pi_s$ from Alg. 1, number of iterations $K$, truncated horizon $H$, confidence level $\delta$, accuracy parameter $\varepsilon$, $L = (1-\gamma)^{-1} + 2\gamma(1-\gamma)^{-2}$.
2: **Initialize:** Set counters $l = 1$, $k_0 = 1$, and $m = 0$, initial mixture weight $\alpha_{l,0} = 0$, $\forall l \in \mathbb{N}$.
3: Select policy with the highest reward value:

$$(\pi_l, u_l, v_l, u_{l,s}, v_{l,s}) = \arg \max_{(\pi,u,v,u_s,v_s)\in\hat{\mathcal{U}}} u$$

4: **for** $k = 1, \ldots, K$ **do**
5:     Set $\pi_k = \pi_{l,m} = \alpha_{l,m}\pi_l + (1 - \alpha_{l,m})\pi_s$ and compute $u_{l,m}, v_{l,m}$ as per Eq. (3).
6:     Sample a trajectory $\tau_H$ using $\pi_{l,m}$ and compute $R_k$ and $C_k$ as per Eq. (4).
7:     **if** Inq. (5) is violated **then**
8:         Update $\hat{\mathcal{U}}$: $\hat{\mathcal{U}} = \hat{\mathcal{U}} \setminus \{(\pi_l, u_l, v_l, u_{l,s}, v_{l,s})\}$
9:         Update counters: $k_0 = k + 1$, $l = l + 1$, $m = 0$
10:         Select policy with the highest reward value:

$$(\pi_l, u_l, v_l, u_{l,s}, v_{l,s}) = \arg \max_{(\pi,u,v,u_s,v_s)\in\hat{\mathcal{U}}} u$$

11:     **else if** $k - k_0 - 1 \geq \frac{32\ln(4K/\delta)}{(1-\gamma)^2\nu_{l,m}^2}$ **and** $m \leq m(l)$, where $m(l)$ is defined in Eq. (7) **then**
12:         Update the mixture weight as per Eq. (6)
13:         Update counters: $k_0 = k + 1$, $m = m + 1$
14:     **end if**
15: **end for**
16: **Return:** Final policy $\pi_{out} = \frac{1}{K}\sum_{k=1}^{K}\pi_k$.

---

At the start of each verification phase, Algorithm 2 selects the policy $\pi_l \in \hat{\mathcal{U}}$ that maximizes the predicted reward $u_l$ (Line 3), following the principle of optimism in the face of uncertainty. Since $\pi_l$ may violate constraints in $\mathcal{M}_{\text{test}}$, the algorithm instead executes the mixture policy $\pi_{l,m} = (1-\alpha_{l,m})\pi_s + \alpha_{l,m}\pi_l$, where the mixture weight is initialized as $\alpha_{l,0} = 0$, which corresponds to executing the feasible policy $\pi_s$ (Line 5). Let $(u_l, v_l)$ and $(u_{l,s}, v_{l,s})$ denote the reward and constraint values of $\pi_l$ and $\pi_s$, respectively, as learned during training and stored in $\hat{\mathcal{U}}$. The predicted reward and constraint values of the mixture policy $\pi_{l,m}$ are then given by the convex combinations

$$u_{l,m} = \alpha_{l,m}u_l + (1 - \alpha_{l,m})u_{l,s}, \tag{3}$$
$$v_{l,m} = \alpha_{l,m}v_l + (1 - \alpha_{l,m})v_{l,s}.$$

To verify whether $\pi_l$ is near-optimal for $\mathcal{M}_{\text{test}}$, the algorithm interacts with $\mathcal{M}_{\text{test}}$ using $\pi_{l,m}$. In Line 6, the algorithm collects a trajectory $\tau_H$ under $\pi_{l,m}$ and computes the corresponding cumulative reward and constraint values:

$$R_k := \sum_{t\in[H-1]} \gamma^t r(s_t, a_t), \quad C_k := \sum_{t\in[H-1]} \gamma^t c(s_t, a_t). \tag{4}$$

By concentration bounds (Ni & Kamgarpour, 2025), the empirical averages of $R_k$ and $C_k$ concentrate around the

true values $(V_r^{\mathcal{M}_{\text{test}}}(\pi_{l,m}), V_c^{\mathcal{M}_{\text{test}}}(\pi_{l,m}))$ with high probability. Furthermore, by Lemma 3.5, if $\pi_l$ is near-optimal for $\mathcal{M}_{\text{test}}$, then the predicted values $(u_{l,m}, v_{l,m})$ are $\mathcal{O}(\varepsilon)$-close to $(V_r^{\mathcal{M}_{\text{test}}}(\pi_{l,m}), V_c^{\mathcal{M}_{\text{test}}}(\pi_{l,m}))$. Consequently, for a near-optimal $\pi_l$, the following condition holds with high probability:

$$\max\left\{\left|\frac{\sum_{j=k_0}^{k} R_j}{k - k_0 + 1} - u_{l,m}\right|, \left|\frac{\sum_{j=k_0}^{k} C_j}{k - k_0 + 1} - v_{l,m}\right|\right\}$$
$$\leq \sqrt{\frac{2\ln(4K/\delta)}{(k - k_0 + 1)(1 - \gamma)^2}} + \varepsilon(L + 1), \tag{5}$$

where the first term captures statistical concentration, and the second term accounts for the estimation error inherited from training.

If the condition in Inequality (5) is violated (Line 7), then $\pi_l$ is not near-optimal for $\mathcal{M}_{\text{test}}$ with high probability and is removed from $\hat{\mathcal{U}}$ (Line 8). The algorithm then resets the verification process by incrementing the policy index $l$, setting the mixture index $m = 0$, and updating the counter $k_0 = k + 1$ (Line 9). This marks the start of a new verification phase for the newly selected policy $\pi_l$ (Line 10).

If $\pi_l$ is not eliminated, the algorithm continues collecting samples. As more data are gathered, the confidence intervals in Inequality (5) shrink, allowing a larger mixture weight $\alpha_{l,m}$ while still maintaining feasibility. In Appendix D, we derive the following closed-form update for $\alpha_{l,m}$ based on the lower confidence bound of the constraint value of $\pi_l$:

$$\alpha_{l,m+1} = \begin{cases} \frac{v_{l,s}-2\varepsilon(L+2)}{v_{l,s}-2\varepsilon(L+2)+2/(1-\gamma)}, & m = 0, \\ \frac{(v_{l,s}-(4L+9)\varepsilon)\alpha_{l,1}}{v_{l,s}\alpha_{l,1}+(v_{l,s}-(4L+9)\varepsilon-v_{l,s}\alpha_{l,1})(C_l)^m}, & m \geq 1, \end{cases} \tag{6}$$

where $C_l = \frac{2}{3} + \frac{4L+9}{3v_{l,s}}$. Note that $C_l$ decreases as $v_{l,s}$ increases, where $v_{l,s}$ denotes the safety margin of the feasible policy $\pi_s$. After updating $\alpha_{l,m}$ (Line 12), the algorithm resets $k_0 = k + 1$ and increments $m$ (Line 13) to begin verifying the updated mixture policy.

The update rule ensures that $\alpha_{l,m}$ converges exponentially fast to $1 - \mathcal{O}(\varepsilon)$, with base $C_l$. Updates stop (Line 11) once

$$m \geq m(l) := \log_{C_l} \varepsilon, \tag{7}$$

at which point the mixture policy differs from $\pi_l$ on only an $\mathcal{O}(\varepsilon)$ fraction of probability mass. Thereafter, the algorithm repeatedly executes $\pi_{l,m}$, achieving near-optimal performance while preserving safety.

In conclusion, the testing phase enables a transition from a conservative yet feasible policy to a high-reward policy, guaranteeing safe exploration. The algorithm returns the final policy as the average mixture of all executed policies (Line 16), and achieves sublinear test-time reward regret, as stated in Theorem 4.2.

# 4. Theoretical Guarantees

We now establish the sample complexity and safe exploration guarantees of Algorithm 2. We begin by introducing a key quantity used to characterize the sample complexity.

**Definition 4.1.** Let $C_\varepsilon(\mathcal{D}, \delta)$ denote the covering number of the highest-density region of $\mathcal{D}$:

$$C_\varepsilon(\mathcal{D}, \delta) := \min\Big\{m \,|\, \exists\, m \text{ CMDPs in } \mathcal{M}_{\text{all}} \text{ denoted by}$$

$$\{\mathcal{M}_i\}_{i \in [m]} \text{ s.t.} \Pr_{\mathcal{M} \sim \mathcal{D}}\Big(\mathbf{1}[\mathcal{M} \in \bigcup_{i=1}^{m} B(\mathcal{M}_i, \varepsilon)]\Big) \geq 1 - \delta\Big\}.$$

The quantity $C_\varepsilon(\mathcal{D}, \delta)$ was considered in (Ye et al., 2023) for countable $\mathcal{M}_{\text{all}}$. Here, we extend it to the general case where $\mathcal{M}_{\text{all}}$ may be uncountable. Intuitively, $C_\varepsilon(\mathcal{D}, \delta)$ measures the minimal number of CMDPs whose $\varepsilon$-neighborhoods cover most of the distribution $\mathcal{D}$ with probability at least $1 - \delta$. Smaller $\varepsilon$ or $\delta$ increases $C_\varepsilon(\mathcal{D}, \delta)$. Even when $|\Omega|$ is large or infinite, $C_\varepsilon(\mathcal{D}, \delta)$ is still finite (see Appendix B) and can be much smaller if $\mathcal{D}$ is concentrated.

To provide more intuition for $C_\varepsilon(\mathcal{D}, \delta)$, consider $\mathcal{D}$ as a $d$-dimensional Gaussian $\mathcal{N}(\mu, \sigma^2 I_d)$. Then, it equals $\mathcal{O}\big(\varepsilon^{-d}(F_{\chi_d^2}^{-1}(1 - \delta))^{d/2}\sigma^d\big)$, where $F_{\chi_d^2}^{-1}(1 - \delta)$ is the $(1 - \delta)$-quantile of the chi-squared distribution with $d$ degrees of freedom. In one dimension, this reduces to $\mathcal{O}(\varepsilon^{-1}\sigma\sqrt{\ln \delta^{-1}})$. For a uniform distribution over a compact set $K \subset \mathbb{R}^d$, $C_\varepsilon(\mathcal{D}, \delta)$ scales as $\mathcal{O}\big(\varepsilon^{-d}(1 - \delta)|K|\big)$, where $|K|$ grows with the dimension $d$.

Using this quantity together with Slater's condition, we obtain the following test-time guarantees for Algorithm 2.

**Theorem 4.2.** *Let Assumption 3.3 hold and set $(8L + 18)\varepsilon \leq \xi$ and the truncated horizon as $H = \tilde{\mathcal{O}}((1 - \gamma)^{-1})$. Then, for any test CMDP $\mathcal{M} \sim \mathcal{D}$, Algorithms 1 and 2 satisfies the following properties with probability at least $1 - 26\delta - 9\delta \ln C_\varepsilon(\mathcal{D}, \delta)$:*

1. *For all $k \in [K]$, each policy $\pi_k$ is feasible. In other words, we have safe exploration (see Definition 2.2).*

2. *The test-time reward regret defined in Equation (1) is bounded by*

$$\text{Reg}_r^{\mathcal{M}}(K) \leq \tilde{\mathcal{O}}\left(\frac{\sqrt{\mathcal{C}_\varepsilon(\mathcal{D}, \delta)K}}{(1 - \gamma)^2\xi} + \frac{\varepsilon K}{\xi(1 - \gamma)^3}\right).$$

The regret bound has two terms: a sublinear term in $K$ scaling with $\mathcal{C}_\varepsilon(\mathcal{D}, \delta)$, and a linear term in $K$. This linear term arises for two reasons. First, the policy learned during training is only $\mathcal{O}(\varepsilon)$-optimal, causing suboptimality to accumulate over $K$ steps, as also observed in meta RL (Ye et al., 2023). Second, estimation errors arising from approximating the reward and constraint values from finite horizon

trajectories further contribute with a term that grows linearly in $K$. This has been commonly observed in RL guarantees (Agarwal et al., 2021; Ding et al., 2020).

Compared with unconstrained meta RL (Ye et al., 2023), the above regret bound preserves the dependence on $K$ and $\mathcal{C}_\varepsilon(\mathcal{D}, \delta)$, while additionally depending on $\xi$ from Slater's condition (Assumption 3.3). Such dependence is typical in constrained RL (Ding et al., 2020; Bura et al., 2022).

Based on this regret bound, we can determine the algorithm's sample complexity required to ensure safe exploration and achieve $\varepsilon$-optimality, as formalized below.

**Corollary 4.3.** *Let Assumption 3.3 hold. For any $\varepsilon, \delta > 0$, Algorithm 2 requires a sample complexity of $\tilde{\mathcal{O}}\Big(\xi^{-2}\varepsilon^{-2}(1 - \gamma)^{-5}\mathcal{C}_{\xi\varepsilon(1-\gamma)^3}(\mathcal{D}, \delta)\Big)$ to ensure that, for any $\mathcal{M} \sim \mathcal{D}$, the output policy is $\mathcal{O}(\varepsilon)$-optimal and feasible for $\mathcal{M}$, and safe exploration is guaranteed with high probability.*

*Remark* 4.4. Without a training stage, the best-known safe exploration results have sample complexities $\tilde{\mathcal{O}}\big(\xi^{-2}\varepsilon^{-2}(1 - \gamma)^{-6}|\mathcal{S}|^2|\mathcal{A}|\big)$ for model-based approaches (Yu et al., 2025) and $\tilde{\mathcal{O}}\big(\min\{\xi^{-6}, \varepsilon^{-6}\}(1 - \gamma)^{-22}\big)$ for model-free approaches (Ni & Kamgarpour, 2025). In contrast, Algorithm 2 leverages prior training knowledge and is more sample efficient when $\mathcal{C}_\varepsilon(\mathcal{D}, \delta)$ is small relative to $|\mathcal{S}|$ and $|\mathcal{A}|$, e.g., for sub-Gaussian distributions, outperforming the best-known methods. When $\mathcal{C}_\varepsilon(\mathcal{D}, \delta)$ is large, such as for a $d$-dimensional uniform distribution scaling as $\varepsilon^{-d}$, training phase offers little benefit.

The full proof of Theorem 4.2 is provided in Appendix E. In the remainder of this section, we provide its proof sketch.

## 4.1. Proof Sketch of Theorem 4.2

Algorithm 1 considers a sequence of policies $\{\pi_l\}_{l=1}^{E} \subset \hat{\mathcal{U}}$, where $E \leq |\hat{\mathcal{U}}|$. For each $\pi_l$, the deployed policy $\pi_{l,m}$ is a mixture of $\pi_l$ and the feasible policy $\pi_s$, with mixture weight $\alpha_{l,m}$ for $m \in [0, m(l)]$. We denote by $\tau_{l,m}$ the first episode in which $\pi_{l,m}$ is used.

We first establish that all mixture policies remain feasible throughout testing, while the value estimates of $\pi_l$ remain optimistically close to the optimal value of the test CMDP.

**Lemma 4.5.** *Assume the events in Lemma 3.5 hold, and set the truncated horizon $H = \tilde{\mathcal{O}}\big((1 - \gamma)^{-1}\big)$. Then, for any $\mathcal{M} \sim \mathcal{D}$, Algorithm 2 guarantees, with probability at least $1 - \delta$, that:*

1. *Safe exploration is ensured during testing.*

2. *At every iteration, the value estimate satisfies*

$$u_l \geq V_r^{\mathcal{M}}(\pi^*) - 4\varepsilon L\big(1 + \xi^{-1}(1 - \gamma)^{-1}\big).$$

The proof of Lemma 4.5 relies on three key ingredients: (i) concentration bounds for the mixture policy, which control the deviation between its empirical and true reward and constraint values; (ii) an inductive argument over iterations showing that each updated mixture policy remains feasible, starting from the feasible policy $\pi_s$; and (iii) the elimination rule in Algorithm 2, which, together with the concentration bounds, guarantees that the near-optimal policy is never removed from the policy-value set. The full proof is provided in Appendix D.

Next, we bound the test-time reward regret by decomposing it into four terms:

$$
\begin{aligned}
\text{Reg}_r^{\mathcal{M}}(K) = \sum_{l=1}^{E} \sum_{m=0}^{m(l)} \sum_{\tau=\tau_{l,m}}^{\tau_{l,m+1}-1} & \Big( \underbrace{V_r^{\mathcal{M}}(\pi^*) - u_l}_{(i)} \\
+ \underbrace{u_l - (\alpha_{l,m} u_l + (1-\alpha_{l,m} u_{l,s}))}_{(ii)} & \\
+ \underbrace{(\alpha_{l,m} u_l + (1-\alpha_{l,m} u_{l,s})) - R_\tau}_{(iii)} + \underbrace{R_\tau - V_r^{\mathcal{M}}(\pi_{l,m})}_{(iv)} \Big).
\end{aligned}
$$

**Term (i):** By Lemma 4.5, the difference between the optimal value $V_r^{\mathcal{M}}(\pi^*)$ and $v_l$ is upper bounded.

**Term (ii):** This term arises from deploying the mixture of $\pi_l$ and $\pi_s$, which ensures safe exploration. It scales as $\mathcal{O}(1 - \alpha_{l,m})$, and since $1 - \alpha_{l,m}$ decreases exponentially toward $\varepsilon$, the term remains small.

**Term (iii):** This term captures the gap between the mixture policy's expected value from the training phase and its empirical value observed during testing. The elimination rule in line 7 of Algorithm 2 discards policies with large deviations, ensuring that this term remains bounded.

**Term (iv):** This term accounts for the estimation error between the empirical and true values of the executed mixture policy. Standard concentration results guarantee that this error remains bounded with high probability.

Summing the four terms gives the test-time reward regret bound, with detailed formulas provided in Appendix E.

## 5. Lower Bound

We continue by establishing a novel lower bound on the test-time sample complexity.

**Theorem 5.1.** *Assume that every CMDP $\mathcal{M} \in \mathcal{M}_{\text{all}}$ has access to a generative model, and that Assumption 3.3 holds. There exists a distribution $\mathcal{D}$ over $\mathcal{M}_{\text{all}}$ and constants $\gamma_0, \varepsilon_0, \delta_0 \in (0,1)$ such that, for any discount factor $\gamma \in (\gamma_0, 1)$, $\varepsilon \in (0, \varepsilon_0)$, and $\delta \in (0, \delta_0)$, any algorithm requires at least $\tilde{\Omega}\Big( \xi^{-2} \varepsilon^{-2} (1-\gamma)^{-5} \mathcal{C}_{\xi\varepsilon(1-\gamma)^3}(\mathcal{D}, \delta) \Big)$ samples from the generative model to guarantee that, for any*

$\mathcal{M} \sim \mathcal{D}$, *the returned policy is both $\varepsilon$-optimal and feasible with probability at least $1 - \delta$.*

The full proof is in Appendix G. Extending the hard-instance construction of (Vaswani et al., 2022) from constrained RL to constrained meta RL, we consider a set of CMDPs $\mathcal{M}_{\text{all}}$ that differ in transition dynamics. The agent must identify each CMDP in $\mathcal{M}_{\text{all}}$ and learn its optimal feasible policy, which leads to a lower bound on the sample complexity. This lower bound provides two key insights. First, it matches the upper bound in Corollary 4.3, showing that the sample complexity of Algorithm 2 is near-optimal up to logarithmic factors. Second, compared with lower bounds for meta RL that do not impose feasibility requirements on the optimal policy (Ye et al., 2023), our bound depends on $\xi$, which measures the safety margin of the initial policy. As $\xi$ decreases, the problem becomes harder in terms of sample complexity.

## 6. Computational Experiment

We empirically evaluate our algorithms in a $7 \times 7$ gridworld adapted from (Sutton & Barto, 2018). We compare against three baselines that ensure safe exploration during testing: *Safe Meta-RL* (Xu & Zhu, 2026) from the constrained meta RL setting, transferring knowledge from training to testing; and two constrained RL approaches without a training phase, namely DOPE+ (Yu et al., 2025) for model-based and LB-SGD (Ni & Kamgarpour, 2025) for model-free.

For each gridworld CMDP $\mathcal{M}_i$, the objective is to reach a rewarded terminal cell (green rectangle) while limiting the cumulative time spent in unsafe regions (red rectangles). The environment consists of four actions: *up*, *right*, *down*, and *left*. The transition dynamics are as follows: with probability $1 - i$, the intended action is executed, and with probability $\frac{i}{4}$, a random action is taken, where $i$ denotes the environment noise. The constraints are defined as follows: entering red rectangles in the third and fifth columns incurs a cost of 10. Upon reaching the goal, the agent remains there and receives a reward of 10 at each step. We set $\gamma = 0.9$ and solve the following constrained problem:

$$
\max_{\pi} V_r^{\mathcal{M}_i}(\pi) \quad \text{s.t.} \quad V_c^{\mathcal{M}_i}(\pi) \leq 1.5.
$$

Here, we consider the family of CMDPs $\mathcal{M}_{\text{all}} = \{\mathcal{M}_i\}_{i \in [0, 0.5]}$, where each CMDPs differ only in their noise level $i$. The parameter $i$ is sampled from a Gaussian distribution $\mathcal{N}(0.3, 0.03)$, truncated to $[0, 0.5]$. When the noise level is low, the optimal policy takes a shorter route directly toward the goal via the fourth column. As the noise increases, this shortcut becomes riskier due to a higher probability of entering unsafe regions, and the optimal policy instead follows a longer but safer path along the boundary of the gridworld. Figure 1 shows the gridworld environment

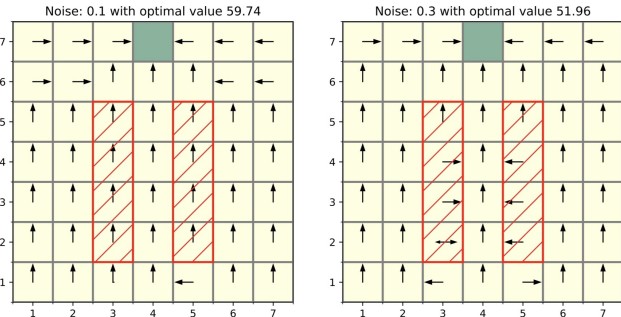

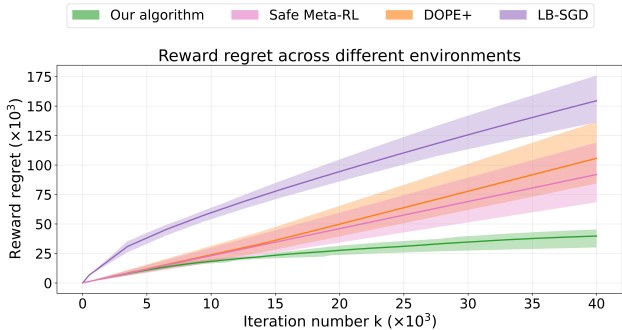

Figure 1: Optimal policies and values for noise levels $i = 0.1, 0.3$.

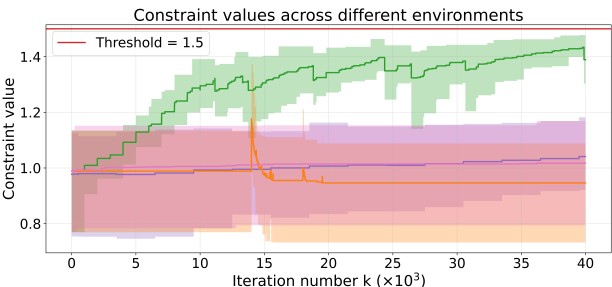

Figure 2: Reward regret and constraint values of our algorithm, Safe Meta-RL, DOPE+, and LB-SGD, averaged over 10 independent runs, where in each run a CMDP $\mathcal{M}_i$ is sampled. All plots share the same x-axis representing the iteration number.

and illustrates the optimal policies along with their corresponding reward values for different noise levels. During the training phase, we use Algorithm 1 to learn a feasible policy $\pi_s$ and the policy-value set $\hat{\mathcal{U}}$. The specific oracles used in Algorithm 1 are detailed in Appendix H.1. We then evaluate Algorithm 2 and all three baselines on 10 test CMDPs, where each CMDP is independently sampled from the distribution. For fair comparison, all algorithms start from the same feasible policy $\pi_s$ learned from training. As shown in Figure 2, all methods ensure safe exploration during testing. Our algorithm achieves roughly 50% lower test-time reward regret than the best-performing baseline. These empirical results align with our theoretical guarantees and demonstrate the benefit of leveraging prior knowledge collected during training.

In addition to the gridworld experiments, we further evaluate our method on continuous locomotion tasks in Gym environments; detailed results are provided in Appendix H.2.

## 7. Discussion and Conclusion

In this paper, we proposed an algorithm for constrained meta RL with provable test-time safety. We established its convergence to an optimal policy and characterized its sample complexity, with a matching problem-dependent lower bound. Empirically, our approach outperforms existing constrained RL and safe meta RL methods in reward performance across gridworld navigation tasks.

Our analysis relies on the linearity of expected reward and constraint values under episodic mixture policies, and thus does not directly extend to nonlinear constraints such as CVaR. Nevertheless, these can be reformulated as expected cumulative constraints via state augmentation (Zhao et al., 2024), under which our results apply. Future work further includes studying constrained meta RL under distribution shift and improving sample complexity by exploiting structural properties of the CMDP set (Mutti & Tamar, 2024).

## Acknowledgment

This work was supported by the Swiss National Science Foundation under Grant 207984.

## Impact Statement

This paper presents work whose goal is to advance the field of machine learning. There are many potential societal consequences of our work, none which we feel must be specifically highlighted here.

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

# Appendix Contents

## A. Comparison of Constrained RL Algorithms

In this section, we compare the sample complexity of learning an $\varepsilon$-optimal policy with safe exploration guarantees against prior constrained RL algorithms, including both model-based and model-free approaches. The results are summarized in Table 1. In the table, $|\mathcal{S}|$ and $|\mathcal{A}|$ denote the cardinalities of the state and action spaces, respectively, $\gamma$ is the discount factor, and $\xi$ is the parameter from Slater's condition, which represents the safety margin of the feasible policy used to initialize learning. For finite-horizon settings, the sample complexity depends explicitly on the horizon $H$, since learning must account for reward and constraint accumulation over $H$ steps. In infinite-horizon discounted problems, this role is played by the effective horizon $(1 - \gamma)^{-1}$, which characterizes how far into the future rewards and constraints significantly influence the return. Consequently, sample complexity bounds in the infinite-horizon setting scale with $(1 - \gamma)^{-1}$, which plays an analogous role to the horizon length $H$ in finite-horizon analyses (Altman, 2021).

Table 1: Sample complexity for learning an $\varepsilon$-optimal policy with safe exploration guarantees with high probability.

| Constrained RL approach | Sample complexity |
|---|---|
| OptPess-LP (Liu et al., 2021) | $\tilde{\mathcal{O}}(\xi^{-2}\varepsilon^{-2}(1 - \gamma)^{-7}|\mathcal{S}|^3|\mathcal{A}|)$ |
| DOPE (Bura et al., 2022) | $\tilde{\mathcal{O}}(\xi^{-2}\varepsilon^{-2}(1 - \gamma)^{-7}|\mathcal{S}|^2|\mathcal{A}|)$ |
| DOPE+ (Yu et al., 2025) | $\tilde{\mathcal{O}}(\xi^{-2}\varepsilon^{-2}(1 - \gamma)^{-6}|\mathcal{S}|^2|\mathcal{A}|)$ |
| LB-SGD (Ni & Kamgarpour, 2025) | $\tilde{\mathcal{O}}((1 - \gamma)^{-22} \min\{\xi^{-6}, \varepsilon^{-6}\})$ |
| **Constrained meta RL approach** | **Sample complexity** |
| This work | $\tilde{\mathcal{O}}(\xi^{-2}\varepsilon^{-2}(1 - \gamma)^{-5}\mathcal{C}_\varepsilon(\mathcal{D}, \delta))$ |

## B. Boundedness of $C_\varepsilon(\mathcal{D}, \delta)$

**Proposition B.1.** *Let $\mathcal{D}$ be a probability distribution over an index set $\Omega \subset \mathbb{R}^d$, and let $f : \Omega \to \mathcal{M}_{\text{all}}$ be a continuous mapping from $\Omega$ to the family of CMDPs $\mathcal{M}_{\text{all}}$. For any $\varepsilon > 0$ and $\delta \in (0, 1)$, define*

$$C_\varepsilon(\mathcal{D}, \delta) := \min \left\{ m \;\middle|\; \exists \{\mathcal{M}_i\}_{i=1}^m \subset \mathcal{M}_{\text{all}} \text{ such that } \Pr_{i^* \sim \mathcal{D}} \left( \mathbf{1}[\mathcal{M}_{i^*} \in \bigcup_{i=1}^m B(\mathcal{M}_i, \varepsilon)] \right) \geq 1 - \delta \right\}.$$

*Then $C_\varepsilon(\mathcal{D}, \delta)$ is finite.*

*Proof.* Since $\mathbb{R}^d$ is a Polish space, every probability measure on it is tight (Bogachev, 2007, Theorem 7.1.7). That is, for any $\delta > 0$, there exists a compact set $K \subset \Omega \subset \mathbb{R}^d$ such that

$$\Pr_{i^* \sim \mathcal{D}}(\mathbf{1}[i^* \in K]) \geq 1 - \delta.$$

Since continuous images of compact sets are compact, the set

$$\tilde{K} := \{\mathcal{M}_i \mid i \in K\} = f(K) \subset \mathcal{M}_{\text{all}}$$

is compact in $\mathcal{M}_{\text{all}}$. Therefore, for any $\varepsilon > 0$, there exists a finite set $\{\mathcal{M}_i\}_{i=1}^m \subset \tilde{K}$ such that

$$\tilde{K} \subset \bigcup_{i=1}^m B(\mathcal{M}_i, \varepsilon),$$

It follows that

$$\Pr_{i^* \sim \mathcal{D}} \left( \mathbf{1}[\mathcal{M}_{i^*} \in \bigcup_{i=1}^m B(\mathcal{M}_i, \varepsilon)] \right) \geq \Pr_{i^* \sim \mathcal{D}}(\mathbf{1}[f(i^*) \in \tilde{K}]) = \Pr_{i^* \sim \mathcal{D}}(\mathbf{1}[i^* \in K]) \geq 1 - \delta.$$

By the definition of $C_\varepsilon(\mathcal{D}, \delta)$, the above inequality shows that $C_\varepsilon(\mathcal{D}, \delta)$ is finite. $\square$

## C. Training Phase

### C.1. Subroutine for Finding a CMDP Set

In this section, we introduce Subroutine 3, which is adapted from (Ye et al., 2023). While (Ye et al., 2023) identifies a set of *policies*, our modification instead identifies a set of *CMDPs*, denoted by $\mathcal{U}$. Specifically, given $N$ sampled CMDPs $\{\mathcal{M}_i\}_{i \in [N]}$, the subroutine ensures that at least a $(1 - 3\delta)$ fraction of $\{\mathcal{M}_i\}_{i \in [N]}$ lies within an $\varepsilon$-neighborhood of $\mathcal{U}$, that is,

$$\sum_{i=1}^{N} \mathbf{1}[\exists \, \mathcal{M} \in \mathcal{U} \text{ such that } \mathcal{M}_i \in B(\mathcal{M}, \varepsilon)] \geq (1 - 3\delta)N.$$

Subroutine 3 greedily selects these CMDPs in at most $N$ iterations. First, we construct a binary matrix $\mathbf{A} \in \mathbb{N}^{N \times N}$, where each entry

$$\mathbf{A}_{i,j} := \mathbb{O}_c(\mathcal{M}_i, \mathcal{M}_j, \varepsilon, \delta/N^2)$$

indicates whether $\mathcal{M}_i$ lies within an $\varepsilon$-neighborhood of $\mathcal{M}_j$. Compared with (Ye et al., 2023, Subroutine 4), our *only* modification is the definition of the matrix $\mathbf{A}$: each entry here measures the similarity between *CMDPs* rather than between *policies*.

At each iteration $t$, the algorithm maintains two sets: a cover set $\mathcal{U}_t$, containing the indices of the selected CMDPs, and an uncovered set $\mathcal{T}_t$, containing the indices of CMDPs that are not within the $\varepsilon$-neighborhood of any CMDP in $\mathcal{U}_t$. We initialize $\mathcal{T}_0 = [N]$ and $\mathcal{U}_0 = \emptyset$. To update these sets, the algorithm selects the CMDP $\mathcal{M}_{j_t}$ whose $\varepsilon$-neighborhood contains the largest number of CMDPs in the current uncovered set $\mathcal{T}_{t-1}$. The selected index is then added to the cover set $\mathcal{U}_t$, and all CMDPs lying within the $\varepsilon$-neighborhood of $\mathcal{M}_{j_t}$ are removed from the uncovered set to form $\mathcal{T}_t$. Subroutine 3 terminates once the size of uncovered index set $\mathcal{T}_t$ satisfies

$$|\mathcal{T}_t| \leq 3\delta N,$$

that is, when at least a $(1 - 3\delta)$ fraction of the sampled CMDPs lie within the $\varepsilon$-neighborhood of the CMDPs indexed by $\mathcal{U}_t$. The algorithm then returns the resulting CMDP set $\mathcal{U} := \{\mathcal{M}_i\}_{i \in \mathcal{U}_t}$.

---

**Algorithm 3** Finding a CMDP set modified from (Ye et al., 2023, Subroutine 4)

---

1: **Input:** $N$ sampled CMDPs $\{\mathcal{M}_i\}_{i \in [N]}$
2: **Initialize:** $\mathcal{U}_0 = \emptyset$ (cover index set), $\mathcal{T}_0 = [N]$ (uncovered index set)
3: Construct a matrix $\mathbf{A} \in \{0, 1\}^{N \times N}$, where $\mathbf{A}_{i,j} = \mathbb{O}_c(\mathcal{M}_i, \mathcal{M}_j, \varepsilon, \delta/N^2)$
4: **for** $t = 1, \ldots, N$ **do**
5:     Select an index: $j_t = \arg\max_{j \in [N] \setminus \mathcal{U}_{t-1}} \sum_{i \in \mathcal{T}_{t-1}} \mathbf{A}_{i,j}$.
6:     Update the sets: $\mathcal{U}_t = \mathcal{U}_{t-1} \cup \{j_t\}$, $\mathcal{T}_t = \mathcal{T}_{t-1} \setminus \{i \in \mathcal{T}_{t-1}, \mathbf{A}_{i,j_t} = 1\}$.
7:     **if** $|\mathcal{T}_t| \leq 3\delta N$ **then**
8:         Construct a CMDP set $\mathcal{U} := \{\mathcal{M}_i\}_{i \in \mathcal{U}_t}$ and **break**
9:     **end if**
10: **end for**
11: **Output:** A CMDP set $\mathcal{U}$

---

### C.2. Proof of Lemma 3.5

In this section, we prove Lemma 3.5 below.

*Proof of Lemma 3.5.* Let $\Omega^* := \bigcup_{i=1}^{C_\varepsilon(\mathcal{D}, \delta)} \mathcal{M}_i^*$ denote a set of CMDPs whose $\varepsilon$-neighborhoods cover the entire task family $\mathcal{M}_{\text{all}}$ with probability at least $1 - \delta$, i.e.,

$$\Pr_{\mathcal{M} \sim \mathcal{D}} \left[ \mathbf{1}[\mathcal{M} \in \Omega^*] \right] \geq 1 - \delta.$$

At each iteration, we rely on the following two auxiliary results, whose proofs are provided in Appendix C.3.

**Lemma C.1.** *For each iteration of Algorithm 1, in which $N$ CMDPs are sampled from $\mathcal{D}$ at the beginning of the iteration, the following hold with probability at least $1 - 3\delta$:*

1. $\left| \frac{1}{N} \sum_{i=1}^{N} \mathbf{1}[\mathcal{M}_i \in \Omega^*] - \Pr_{\mathcal{M} \sim \mathcal{D}}[\mathcal{M} \in \Omega^*] \right| \leq \delta$.

2. *For all $i, j \in [N]$, the return of oracle $\mathbb{O}_c(\mathcal{M}_i, \mathcal{M}_j, \varepsilon, \delta/N^2)$ equals $\mathbf{1}(\mathcal{M}_i \in B(\mathcal{M}_j, \varepsilon))$.*

3. *For any index set $\mathcal{U}' \subset [N]$,*

$$\Pr_{\mathcal{M} \sim \mathcal{D}}[\exists\, j \in \mathcal{U}', \, \mathcal{M} \in B(\mathcal{M}_j, \varepsilon)] \geq \frac{1}{N - |\mathcal{U}'|} \sum_{i \in [N] \setminus \mathcal{U}'} \max_{j \in \mathcal{U}'} \mathbf{A}_{i,j} - 2\delta - \sqrt{\frac{|\mathcal{U}'| \ln(2N/\delta)}{N - |\mathcal{U}'|}}.$$

The proof of the above lemma is provided in Appendix C.3.1.

**Lemma C.2.** *If the first two events in Lemma C.1 hold, then the size of $\mathcal{U}$ returned by Subroutine 3 satisfies*

$$|\mathcal{U}| \leq (C_\varepsilon(\mathcal{D}, \delta) + 1) \ln(1/\delta).$$

We provide the proof of Lemma C.2 in Appendix C.3.2. Using the results of Lemma C.1 together with the termination condition of Algorithm 1, which stops sampling CMDPs once

$$\sqrt{\frac{|\mathcal{U}| \ln(2N/\delta)}{N - |\mathcal{U}|}} \leq \delta,$$

we see from Lemma C.2 that this condition is satisfied if the total sample size $N$ is lower bounded by

$$N \geq N_0 := \frac{4 C_\varepsilon(\mathcal{D}, \delta) \ln^2\left(C_\varepsilon(\mathcal{D}, \delta)/\delta\right)}{\delta^2}.$$

Since the sample size doubles at each iteration in Algorithm 1, starting from the initial size $\frac{\ln^2(\delta)}{\delta^2}$, the total number of iterations required by Algorithm 1 is at most

$$
\begin{aligned}
\ln \frac{2\delta^2 N_0}{\ln^2(\delta)} &= \ln \frac{8 C_\varepsilon(\mathcal{D}, \delta) \ln^2(C_\varepsilon(\mathcal{D}, \delta)/\delta)}{\ln^2(\delta)} \\
&\overset{(i)}{\leq} \ln\left( \frac{16 C_\varepsilon(\mathcal{D}, \delta)\left(\ln^2 C_\varepsilon(\mathcal{D}, \delta) + \ln^2(\delta)\right)}{\ln^2(\delta)} \right) \\
&= 4 + \ln C_\varepsilon(\mathcal{D}, \delta) + \ln\left(\log_\delta^2 C_\varepsilon(\mathcal{D}, \delta) + 1\right) \\
&\overset{(ii)}{\leq} 6 + \ln C_\varepsilon(\mathcal{D}, \delta) + 2\ln\left(|\log_\delta C_\varepsilon(\mathcal{D}, \delta)|\right) \\
&\overset{(iii)}{\leq} 6 + 3\ln C_\varepsilon(\mathcal{D}, \delta).
\end{aligned}
$$

where (i) uses $\ln^2(xy) \leq 2\ln^2 x + 2\ln^2 y$ for any $x, y > 0$, (ii) uses $\ln(1 + x^2) \leq 2\ln 2 + 2\ln x = 2 + 2\ln x$ for any $x > 0$, (iii) follows from $\delta < 1$ which implies $|\log_\delta C_\varepsilon(\mathcal{D}, \delta)| \leq C_\varepsilon(\mathcal{D}, \delta)$. Meanwhile, the total number of sampled CMDPs is bounded by

$$2N_0 = \tilde{\mathcal{O}}\left( \frac{C_\varepsilon(\mathcal{D}, \delta)}{\delta^2} \right).$$

By a union bound over all iterations, events in Lemmas C.1 and C.2 hold simultaneously with probability at least $1 - 18\delta - 9\delta \ln C_\varepsilon(\mathcal{D}, \delta)$. In the remainder of the proof, we condition on the event that Lemmas C.1 and C.2 hold.

Applying the third event of Lemma C.1 on the final set $\mathcal{U}$ gives

$$
\begin{aligned}
\Pr_{\mathcal{M} \sim \mathcal{D}}[\exists\, \mathcal{M}_j \in \mathcal{U}, \, \mathcal{M} \in B(\mathcal{M}_j, \varepsilon)] &\geq \frac{1}{N - |\mathcal{U}|} \sum_{i \in [N] \setminus \mathcal{U}} \max_{j \in \mathcal{U}} \mathbf{A}_{i,j} - 2\delta - \sqrt{\frac{|\mathcal{U}| \ln(2N/\delta)}{N - |\mathcal{U}|}} \\
&\geq \frac{(1 - 3\delta)N - |\mathcal{U}|}{N - |\mathcal{U}|} - 2\delta - \sqrt{\frac{|\mathcal{U}| \ln(2N/\delta)}{N - |\mathcal{U}|}}
\end{aligned}
$$

$$\geq 1 - 6\delta, \tag{8}$$

where the second inequality follows from the termination condition of Subroutine 3, the last uses the stopping criterion of Algorithm 1, and $|\mathcal{U}| \leq N$.

Given the final CMDP set $\mathcal{U}$, we apply:

- Oracle $\mathbb{O}_l(\mathcal{M}_i, \varepsilon, \frac{\delta}{2|\mathcal{U}|})$ to obtain $\varepsilon$-optimal, relaxed $\varepsilon$-feasible policies $\pi_i$ for each $\mathcal{M}_i \in \mathcal{U}$.

- Oracle $\mathbb{O}_s(\mathcal{U}, \xi, \frac{\delta}{2})$ to obtain a $\xi$-feasible policy $\pi_s$ for all $\mathcal{M}_i \in \mathcal{U}$.

By a union bound, all oracle guarantees hold with probability at least $1 - \delta$. We then construct

$$\hat{\mathcal{U}} := \big\{ (\pi_i, V_r^{\mathcal{M}_i}(\pi_i), V_c^{\mathcal{M}_i}(\pi_i), V_r^{\mathcal{M}_i}(\pi_s), V_c^{\mathcal{M}_i}(\pi_s)) \mid \mathcal{M}_i \in \mathcal{U} \big\}.$$

To evaluate the reward and constraint values of the policies in $\hat{\mathcal{U}}$ and of $\pi_s$ over the entire task family $\mathcal{M}_{\text{all}}$, we rely on the following lemmas, with proofs provided in Appendix C.3.

**Lemma C.3.** *For any policy $\pi \in \Pi$ and any $\mathcal{M}_i, \mathcal{M}_j \in \mathcal{M}_{\text{all}}$,*

$$\max\big\{ \big|V_r^{\mathcal{M}_i}(\pi) - V_r^{\mathcal{M}_j}(\pi)\big|, \big|V_c^{\mathcal{M}_i}(\pi) - V_c^{\mathcal{M}_j}(\pi)\big| \big\} \leq L \, d(\mathcal{M}_i, \mathcal{M}_j),$$

*where $L = \frac{1}{1-\gamma} + \frac{2\gamma}{(1-\gamma)^2}$.*

This follows from the simulation Lemma (Abbeel & Ng, 2005; Puterman, 2014).

**Lemma C.4.** *Let Assumption 3.3 hold with $2L\varepsilon \leq \xi$. For any $\mathcal{M} \in \mathcal{M}_{\text{all}}$ and any $\mathcal{M}_i, \mathcal{M}_j \in B(\mathcal{M}, \varepsilon)$, the policy $\pi_i^*$ is $2\varepsilon L\big(1 + \frac{2}{\xi(1-\gamma)}\big)$-optimal in $\mathcal{M}_j$.*

Proof of Lemma C.4 is in Appendix C.3.3. Since Inequality (8) implies that for any $\mathcal{M} \sim \mathcal{D}$, there exists $\mathcal{M}_i \in \mathcal{U}$ such that $\mathcal{M} \in B(\mathcal{M}_i, \varepsilon)$ with probability at least $1 - 6\delta$, and guarantees for oracles $\mathbb{O}_l$ and $\mathbb{O}_s$ hold with probability at least $1 - \delta$. Together with Lemma C.4, we have

$$\Pr_{\mathcal{M} \sim \mathcal{D}} \left[ \mathbf{1} \left[ \exists \mathcal{M}_i \in \mathcal{U} \text{ such that } \big|V_r^{\mathcal{M}}(\pi^*) - V_r^{\mathcal{M}_i}(\pi_i^*)\big| \leq 2\varepsilon L \left( 1 + \frac{2}{\xi(1-\gamma)} \right) \right] \right] \geq 1 - 7\delta.$$

Meanwhile, for any $\mathcal{M} \in B(\mathcal{M}_i, \varepsilon)$, Lemma C.3 gives

$$\max\Big\{ \big|V_r^{\mathcal{M}}(\pi_i) - u_i\big|, \big|V_c^{\mathcal{M}}(\pi_i) - v_i\big|, \big|V_r^{\mathcal{M}}(\pi_s) - u_{i,s}\big|, \big|V_c^{\mathcal{M}}(\pi_s) - v_{i,s}\big| \Big\} \leq \varepsilon L.$$

Combining these results, we have that: with probability at least $1 - 7\delta$, for any $\mathcal{M} \sim \mathcal{D}$ there exists $(\pi_i, u_i, v_i, u_{i,s}, v_{i,s}) \in \hat{\mathcal{U}}$ such that

$$\max\Big\{ \big|V_r^{\mathcal{M}}(\pi_i) - u_i\big|, \big|V_c^{\mathcal{M}}(\pi_i) - v_i\big|, \big|V_r^{\mathcal{M}}(\pi_s) - u_{i,s}\big|, \big|V_c^{\mathcal{M}}(\pi_s) - v_{i,s}\big| \Big\} \leq \varepsilon L,$$

$$\big|u_i - V_r^{\mathcal{M}}(\pi^*)\big| \leq \underbrace{\big|u_i - V_r^{\mathcal{M}_i}(\pi_i^*)\big|}_{\leq \varepsilon L} + \underbrace{\big|V_r^{\mathcal{M}_i}(\pi_i^*) - V_r^{\mathcal{M}}(\pi^*)\big|}_{\leq 2\varepsilon L\big(1 + \frac{2}{\xi(1-\gamma)}\big)} \leq 4\varepsilon L \left( 1 + \frac{1}{\xi(1-\gamma)} \right).$$

This completes the proof. $\qquad \square$

### C.3. Proof of Auxiliary Lemmas

In this section, we provide the proofs of Lemmas C.1, C.2, and C.4, which serve as auxiliary results for proving Lemma 3.5.

### C.3.1. PROOF OF LEMMA C.1

*Proof of Lemma C.1.* Since $\mathcal{M}_i \sim \mathcal{D}$, the expectation of the indicator random variable $\mathbf{1}[\mathcal{M}_i \in \Omega^*]$ equals $\Pr_{\mathcal{M} \sim \mathcal{D}}[\mathbf{1}[\mathcal{M} \in \Omega^*]]$. By the Chernoff bound,

$$\Pr\left[\left|\frac{1}{N}\sum_{i=1}^{N}\mathbf{1}[\mathcal{M}_i \in \Omega^*] - \Pr_{\mathcal{M} \sim \mathcal{D}}[\mathbf{1}[\mathcal{M} \in \Omega^*]]\right| > \delta\right] < \exp(-2N\delta^2) < \delta.$$

Next, for any pair $i, j \in [N]$, the event $\mathbb{O}_c(\mathcal{M}_i, \mathcal{M}_j, \varepsilon, \delta/N^2) \neq \mathbf{1}(\mathcal{M}_i \in B(\mathcal{M}_j, \varepsilon))$ occurs with probability at most $\delta/N^2$. Applying a union bound over all pairs $(i, j)$, we obtain that with probability at least $1 - \delta$,

$$\mathbb{O}_c(\mathcal{M}_i, \mathcal{M}_j, \varepsilon, \delta/N^2) = \mathbf{1}(\mathcal{M}_i \in B(\mathcal{M}_j, \varepsilon)), \qquad \forall i, j \in [N].$$

Now fix any index set $\mathcal{U}' \subset [N]$ and let $\mathcal{U}^c = [N] \setminus \mathcal{U}'$. For any CMDPs $\mathcal{M}, \mathcal{M}' \in \Omega$, define

$$\chi(\mathcal{M}, \mathcal{M}') := \mathbb{O}_c(\mathcal{M}, \mathcal{M}', \varepsilon, \delta/N^2).$$

Note that $\mathbf{A}_{i,j} = \chi(\mathcal{M}_i, \mathcal{M}_j)$. For a fixed index set $\mathcal{U}'$, $\{\mathcal{M}_i\}_{i \in \mathcal{U}^c}$ are i.i.d. samples from $\mathcal{D}$. By the Chernoff bound, with probability at least $1 - \frac{\delta}{(2N)^{|\mathcal{U}'|}}$, we have

$$\frac{1}{|\mathcal{U}^c|}\sum_{i \in \mathcal{U}^c} \max_{j \in \mathcal{U}'} \mathbf{A}_{i,j} \leq \mathbb{E}_{\mathcal{M} \sim \mathcal{D}}\left[\max_{j \in \mathcal{U}'} \chi(\mathcal{M}, \mathcal{M}_j)\right] + \sqrt{\frac{|\mathcal{U}'| \ln \frac{2N}{\delta}}{|\mathcal{U}^c|}}. \tag{9}$$

On the other hand,

$$\mathbb{E}_{\mathcal{M} \sim \mathcal{D}}\left[\max_{j \in \mathcal{U}'} \chi(\mathcal{M}, \mathcal{M}_j)\right] = \int_{\Omega} \mathcal{P}(d\mathcal{M}) \Pr\left[\exists j \in \mathcal{U}' : \mathbb{O}_c(\mathcal{M}, \mathcal{M}_j, \varepsilon, \delta/N^2) = 1\right].$$

With probability at least $1 - \delta$, for all $j \in \mathcal{U}'$,

$$\mathbb{O}_c(\mathcal{M}, \mathcal{M}_j, \varepsilon, \delta/N^2) = \mathbf{1}(\mathcal{M} \in B(\mathcal{M}_j, \varepsilon)).$$

Therefore,

$$\mathbb{E}_{\mathcal{M} \sim \mathcal{D}}\left[\max_{j \in \mathcal{U}'} \chi(\mathcal{M}, \mathcal{M}_j)\right] \leq \delta + \Pr_{\mathcal{M} \sim \mathcal{D}}[\exists j \in \mathcal{U}' : \mathcal{M} \in B(\mathcal{M}_j, \varepsilon)]. \tag{10}$$

Combining Inequalities (9) and (10), and applying a union bound over all $\mathcal{U}' \subset [N]$, we obtain that with probability at least

$$1 - \sum_{l=1}^{N} \frac{\delta}{(2N)^l}\binom{N}{l} \geq 1 - \sum_{l=1}^{N} \frac{\delta}{(2N)^l}N^l \geq 1 - \delta,$$

the following holds for all $\mathcal{U}' \subset [N]$:

$$\Pr_{\mathcal{M} \sim \mathcal{D}}[\exists j \in \mathcal{U}' : \mathcal{M} \in B(\mathcal{M}_j, \varepsilon)] \geq \frac{1}{N - |\mathcal{U}'|}\sum_{i \in [N] \setminus \mathcal{U}'} \max_{j \in \mathcal{U}'} \mathbf{A}_{i,j} - 2\delta - \sqrt{\frac{|\mathcal{U}'| \ln(2N/\delta)}{N - |\mathcal{U}'|}}.$$

$\square$

### C.3.2. PROOF OF LEMMA C.2

*Proof of Lemma C.2.* At iteration $t$, define $N_{\mathcal{M},t} = \sum_{i \in \mathcal{T}_t} \mathbf{1}[\mathcal{M}_i \in B(\mathcal{M}, \varepsilon)]$ as the number of CMDPs in the current set $\mathcal{T}_t$ that fall inside the $\varepsilon$-neighborhood of $\mathcal{M}$. We further define $\hat{C} \triangleq \sum_{i=1}^{N} \mathbf{1}[\mathcal{M}_i \in \Omega^*]$. Since $\Omega^* = \bigcup_{i=1}^{C_\varepsilon(\mathcal{D},\delta)} B(\mathcal{M}_i^*, \varepsilon)$, it follows that $\hat{C} \leq \sum_{i=1}^{C_\varepsilon(\mathcal{D},\delta)} N_{\mathcal{M}_i^*, 0}$. By the first event in Lemma C.1, we have

$$\left|\frac{\hat{C}}{N} - \Pr_{\mathcal{M} \sim \mathcal{D}}[\mathbf{1}[\mathcal{M} \in \Omega^*]]\right| \leq \delta,$$

and since $\Pr_{\mathcal{M}\sim\mathcal{D}}[\mathbf{1}[\mathcal{M}\in\Omega^*]] \geq 1-\delta$, this implies $\frac{\hat{C}}{N} \geq 1-2\delta$. Consequently,

$$\sum_{i=1}^{C_\varepsilon(\mathcal{D},\delta)} N_{\mathcal{M}_i^*,0} \geq N(1-2\delta). \tag{11}$$

Note that $\mathcal{U}$ is generated by greedily selecting the CMDP that covers the largest number of remaining CMDPs in $\mathcal{T}_{t-1}$. Moreover, since the second event in Lemma C.1 holds, we have that if $\mathcal{M}_i \in B(\mathcal{M}_j,\varepsilon)$, then $\mathcal{M}_j \in B(\mathcal{M}_i,\varepsilon)$, implying $\mathbf{A}_{i,j} = \mathbf{A}_{j,i} = 1$. For each step $t$ and any $\mathcal{M}_j^* \in \Omega^*$ where $j \in [C_\varepsilon(\mathcal{D},\delta)]$, if $\mathcal{M}_i = \mathcal{M}_j^*$ for some $i \in \mathcal{T}_{t-1}$, then in this step we have

$$\sum_{i'\in\mathcal{T}_{t-1}} \mathbf{A}_{i',i} \geq N_{\mathcal{M}_j^*,t-1}.$$

Since we select $\mathcal{M}_{j_t}$ according to the greedy rule, it must hold that

$$\sum_{i'\in\mathcal{T}_{t-1}} \mathbf{A}_{i',j_t} \geq \sum_{i'\in\mathcal{T}_{t-1}} \mathbf{A}_{i',i} \geq N_{\mathcal{M}_j^*,t-1}, \qquad \forall j \in [C_\varepsilon(\mathcal{D},\delta)].$$

Summing over all $\mathcal{M}_j^*$, we obtain

$$\begin{aligned}
\sum_{i'\in\mathcal{T}_{t-1}} \mathbf{A}_{i',j_t} &\geq \frac{1}{C_\varepsilon(\mathcal{D},\delta)} \sum_{j=1}^{C_\varepsilon(\mathcal{D},\delta)} N_{\mathcal{M}_j^*,t-1} \\
&= \frac{1}{C_\varepsilon(\mathcal{D},\delta)} \left( \sum_{j=1}^{C_\varepsilon(\mathcal{D},\delta)} N_{\mathcal{M}_j^*,0} - \sum_{\mathcal{M}\in\Omega^*} \left(N_{\mathcal{M}_j^*,0} - N_{\mathcal{M}_j^*,t-1}\right) \right) \\
&\overset{(i)}{\geq} \frac{1}{C_\varepsilon(\mathcal{D},\delta)} \left( \sum_{j=1}^{C_\varepsilon(\mathcal{D},\delta)} N_{\mathcal{M}_j^*,0} - (N - |\mathcal{T}_t|) \right) \\
&\overset{(ii)}{\geq} \frac{1}{C_\varepsilon(\mathcal{D},\delta)} \left( |\mathcal{T}_t| - 2\delta N \right),
\end{aligned}$$

where $(i)$ follows since the total number $\sum_{\mathcal{M}\in\Omega^*}(N_{\mathcal{M}_j^*,0} - N_{\mathcal{M}_j^*,t-1})$ is upper bounded by the number of CMDPs covered in the first $t$ rounds, and $(ii)$ uses Inequality (11). By definition, $\sum_{i'\in\mathcal{T}_{t-1}} \mathbf{A}_{i',j_t} = |\mathcal{T}_{t-1}| - |\mathcal{T}_t|$, so the above inequality can be written as

$$|\mathcal{T}_t| - 2\delta N \leq \frac{C_\varepsilon(\mathcal{D},\delta)}{C_\varepsilon(\mathcal{D},\delta)+1} \left( |\mathcal{T}_{t-1}| - 2\delta N \right). \tag{12}$$

If there exists some $t < (C_\varepsilon(\mathcal{D},\delta)+1)\ln(1/\delta)$ such that $|\mathcal{T}_t| \leq 2\delta N$, then the algorithm terminates and

$$|\mathcal{U}| = |\mathcal{U}_t| \leq (C_\varepsilon(\mathcal{D},\delta)+1)\ln\frac{1}{\delta}.$$

Otherwise, for all such $t$ we have $|\mathcal{T}_t| > 2\delta N$, and applying Inequality (12) recursively gives

$$|\mathcal{T}_t| \leq 2\delta N + N(1-2\delta) \left( \frac{C_\varepsilon(\mathcal{D},\delta)}{C_\varepsilon(\mathcal{D},\delta)+1} \right)^t \leq 2\delta N + N(1-2\delta)e^{-t/(C_\varepsilon(\mathcal{D},\delta)+1)} \leq 3\delta N.$$

Hence, upon termination, the size of $\mathcal{U}$ satisfies

$$|\mathcal{U}| = |\mathcal{U}_t| \leq (C_\varepsilon(\mathcal{D},\delta)+1)\ln\frac{1}{\delta}.$$

$\square$

C.3.3. PROOF OF LEMMA C.4

*Proof of Lemma C.4.* We first consider a surrogate CMDP associated with $\mathcal{M}_j$, in which the threshold for constraint satisfaction is relaxed as follows:

$$\max_{\pi \in \Pi} V_r^{\mathcal{M}_j}(\pi), \quad \text{s.t. } V_c^{\mathcal{M}_j}(\pi) \geq -L\varepsilon. \tag{13}$$

Let $\hat{\pi}$ denote an optimal policy of this relaxed CMDP. Since the above problem is a constrained optimization problem over the state and action occupancy space (Altman, 2021), we can apply standard continuity results for optimal objective values in convex optimization. Because Assumption 3.3 holds with $2L\varepsilon \leq \xi$, (Tian et al., 2024, Lemma 14) implies

$$V_r^{\mathcal{M}_j}(\hat{\pi}) \leq V_r^{\mathcal{M}_j}(\pi_j^*) + \frac{4L\varepsilon}{\xi(1-\gamma)}. \tag{14}$$

By Lemma C.3, the optimal policy $\pi_i^*$ of $\mathcal{M}_i$ satisfies

$$V_c^{\mathcal{M}_j}(\pi_i^*) \geq V_c^{\mathcal{M}_i}(\pi_i^*) - L\varepsilon \geq -L\varepsilon,$$

which implies that $\pi_i^*$ satisfies the constraint defined in problem (13). Therefore,

$$V_r^{\mathcal{M}_j}(\pi_i^*) \leq V_r^{\mathcal{M}_j}(\hat{\pi}). \tag{15}$$

Combining Inequalities (14) and (15), we obtain

$$V_r^{\mathcal{M}_j}(\pi_i^*) \leq V_r^{\mathcal{M}_j}(\pi_j^*) + \frac{4L\varepsilon}{\xi(1-\gamma)}. \tag{16}$$

Similarly, we can upper bound the reward value of $\pi_j^*$ in $\mathcal{M}_i$ as

$$V_r^{\mathcal{M}_i}(\pi_j^*) \leq V_r^{\mathcal{M}_i}(\pi_i^*) + \frac{4L\varepsilon}{\xi(1-\gamma)}. \tag{17}$$

Applying Lemma C.3 again to bound the reward value of $\pi_i^*$ and $\pi_j^*$ yields

$$V_r^{\mathcal{M}_j}(\pi_i^*) \geq V_r^{\mathcal{M}_i}(\pi_i^*) - L\varepsilon, \; V_r^{\mathcal{M}_j}(\pi_j^*) \geq V_r^{\mathcal{M}_i}(\pi_j^*) - L\varepsilon. \tag{18}$$

Combining Inequalities (17) and (18), we obtain

$$V_r^{\mathcal{M}_j}(\pi_i^*) \geq V_r^{\mathcal{M}_i}(\pi_i^*) - L\varepsilon \geq V_r^{\mathcal{M}_i}(\pi_j^*) - \frac{2L\varepsilon}{\xi(1-\gamma)} - L\varepsilon \geq V_r^{\mathcal{M}_j}(\pi_j^*) - \frac{4L\varepsilon}{\xi(1-\gamma)} - 2L\varepsilon. \tag{19}$$

Putting Inequalities (16) and (19) together, we conclude that

$$V_r^{\mathcal{M}_j}(\pi_i^*) \in \left[ V_r^{\mathcal{M}_j}(\pi_j^*) - \frac{4L\varepsilon}{\xi(1-\gamma)} - 2L\varepsilon, \; V_r^{\mathcal{M}_j}(\pi_j^*) + \frac{4L\varepsilon}{\xi(1-\gamma)} \right],$$

which implies that $\pi_i^*$ is $2L\varepsilon\left(1 + \frac{2}{\xi(1-\gamma)}\right)$-optimal in $\mathcal{M}_j$. $\square$

# D. Proof of Lemma 4.5

*Proof of Lemma 4.5.* In Lemma 3.5, we show that for any $\mathcal{M} \sim \mathcal{D}$, with probability at least $1 - 25\delta - 9\delta \ln C_\varepsilon(\mathcal{D}, \delta)$, there exists a tuple $(\pi, u, v, u_s, v_s) \in \hat{\Pi}$ such that

$$\max\left\{ \left|V_r^{\mathcal{M}}(\pi) - u\right|, \; \left|V_c^{\mathcal{M}}(\pi) - v\right|, \; \left|V_r^{\mathcal{M}}(\pi_s) - u_s\right|, \; \left|V_c^{\mathcal{M}}(\pi_s) - v_s\right| \right\} \leq \varepsilon L, \tag{20}$$

$$\left|u - V_r^{\mathcal{M}}(\pi^*)\right| \leq 4\varepsilon L\left(1 + \frac{1}{\xi(1-\gamma)}\right). \tag{21}$$

In the following analysis, we focus on the case where the events of Lemma 3.5 hold and show that the tuple $(\pi, u, v, u_s, v_s)$ corresponding to sampled CMDP $\mathcal{M}$ is preserved in the policy-value set $\hat{\mathcal{U}}$ with high probability.

**Step 1: Concentration of empirical reward and constraint estimates.** Conditioned on the sampled CMDP $\mathcal{M}$, consider a fixed iteration $k \in [K]$ and the corresponding mixture coefficient $\alpha_{l,m}$. By (Ni & Kamgarpour, 2025, Proposition 3.3), and setting the truncated horizon

$$H = \mathcal{O}\left(\frac{\ln \varepsilon^{-1}}{1 - \gamma}\right),$$

we obtain that for the policy $\pi_{l,m}$ employed in Algorithm 2 at iteration $k \in [K]$,

$$\Pr\left(\left|\frac{1}{k - k_0 + 1}\sum_{\tau=k_0}^{k} R_\tau - \left[\alpha_{l,m}V_r^{\mathcal{M}}(\pi_l) + (1 - \alpha_{l,m})V_r^{\mathcal{M}}(\pi_s)\right]\right| \geq \varepsilon + \sqrt{\frac{2\ln(4K/\delta)}{(k - k_0 + 1)(1 - \gamma)^2}}\right) \leq \frac{\delta}{2K},$$

$$\Pr\left(\left|\frac{1}{k - k_0 + 1}\sum_{\tau=k_0}^{k} C_\tau - \left[\alpha_{l,m}V_c^{\mathcal{M}}(\pi_l) + (1 - \alpha_{l,m})V_c^{\mathcal{M}}(\pi_s)\right]\right| \geq \varepsilon + \sqrt{\frac{2\ln(4K/\delta)}{(k - k_0 + 1)(1 - \gamma)^2}}\right) \leq \frac{\delta}{2K}.$$

Applying a union bound over all $k \in [K]$, we obtain that with probability at least $1 - \delta$, for all $k \in [K]$,

$$\left|\frac{1}{k - k_0 + 1}\sum_{\tau=k_0}^{k} R_\tau - \left[\alpha_{l,m}V_r^{\mathcal{M}}(\pi_l) + (1 - \alpha_{l,m})V_r^{\mathcal{M}}(\pi_s)\right]\right| < \varepsilon + \sqrt{\frac{2\ln(4K/\delta)}{(k - k_0 + 1)(1 - \gamma)^2}}, \tag{22}$$

$$\left|\frac{1}{k - k_0 + 1}\sum_{\tau=k_0}^{k} C_\tau - \left[\alpha_{l,m}V_c^{\mathcal{M}}(\pi_l) + (1 - \alpha_{l,m})V_c^{\mathcal{M}}(\pi_s)\right]\right| < \varepsilon + \sqrt{\frac{2\ln(4K/\delta)}{(k - k_0 + 1)(1 - \gamma)^2}}. \tag{23}$$

**Step 2: Relating estimation with true values.** For the sub-optimal policy $\pi$ corresponding to the sampled CMDP $\mathcal{M}$, Inequality (20) guarantees that the estimated and true value pairs differ by at most

$$\left|\alpha V_r^{\mathcal{M}}(\pi) + (1 - \alpha)V_r^{\mathcal{M}}(\pi_s) - \left[\alpha u + (1 - \alpha)u_s\right]\right| \leq \varepsilon L,$$

$$\left|\alpha V_c^{\mathcal{M}}(\pi) + (1 - \alpha)V_c^{\mathcal{M}}(\pi_s) - \left[\alpha v + (1 - \alpha)v_s\right]\right| \leq \varepsilon L, \tag{24}$$

for any $\alpha \in [0, 1]$. Combining the above with the concentration bounds in Inequalities (22) and (23), we obtain that, with probability at least $1 - \delta$,

$$\left|\frac{1}{k - k_0 + 1}\sum_{\tau=k_0}^{k} R_\tau - \left[\alpha_{l',m'}u + (1 - \alpha_{l',m'})u_s\right]\right| \leq \sqrt{\frac{2\ln(4K/\delta)}{(k - k_0 + 1)(1 - \gamma)^2}} + \varepsilon(L + 1),$$

$$\left|\frac{1}{k - k_0 + 1}\sum_{\tau=k_0}^{k} C_\tau - \left[\alpha_{l',m'}v + (1 - \alpha_{l',m'})v_s\right]\right| \leq \sqrt{\frac{2\ln(4K/\delta)}{(k - k_0 + 1)(1 - \gamma)^2}} + \varepsilon(L + 1),$$

where $\alpha_{l',m'}$ is the weight when policy $\pi$ selected for constructing the mixture policy.

**Step 3: Non-elimination of the optimal tuple.** By the elimination rule (line 7 of Algorithm 2), the above inequalities ensure that the tuple $(\pi, u, v, u_s, v_s)$ will *not* be eliminated from $\hat{\mathcal{U}}$ with probability at least $1 - \delta$. Combining Inequality (21) with the fact that, in each iteration of Algorithm 2, $u_l = \max_{(\pi, u, v, u_s, v_s) \in \hat{\mathcal{U}}} u$, we obtain

$$u_l \geq V_r^{\mathcal{M}}(\pi^*) - 4\varepsilon L\left(1 + \frac{1}{\xi(1 - \gamma)}\right).$$

Next, we show that Algorithm 2 ensures safe exploration. Under the condition in line 11 of Algorithm 2, together with the elimination rule in line 7, we update the weights $\alpha_{l,m}$ if and only if

$$\left|\frac{1}{k - k_0 + 1}\sum_{\tau=k_0}^{k} C_\tau - v_{l,m}\right| \leq \frac{v_{l,m}}{4} + \varepsilon(L + 1),$$

where $k - k_0 + 1 = \frac{32 \ln(4K/\delta)}{(1-\gamma)^2 \nu_{l,m}^2}$. Combining the above inequality with Inequality (23), we have

$$\alpha_{l,m} V_c^{\mathcal{M}}(\pi) + (1 - \alpha_{l,m}) V_c^{\mathcal{M}}(\pi_s) \in \left[ \frac{v_{l,m}}{2} - \varepsilon(L+2), \ \frac{3v_{l,m}}{2} + \varepsilon(L+2) \right]. \tag{25}$$

Here, we will prove safe exploration by induction on $m \in [0, m(l)]$ for any fixed $l$.

**Base case** ($m = 0, 1$). For $m = 0$, $\pi_{l,0} = \pi_s$ is $(\xi - L\varepsilon)$-feasible by Lemma 3.5. By Inequality (25) with $m = 0$, we have

$$\alpha_{l,0} V_c^{\mathcal{M}}(\pi) + (1 - \alpha_{l,0}) V_c^{\mathcal{M}}(\pi_s) = V_c^{\mathcal{M}}(\pi_s) \in \left[ \frac{v_{l,s}}{2} - \varepsilon(L+2), \ \frac{3v_{l,s}}{2} + \varepsilon(L+2) \right]. \tag{26}$$

Therefore, the feasibility of the next mixture policy $\pi_{l,1}$ follows from

$$V_c^{\mathcal{M}}(\pi_{l,1}) = \alpha_{l,1} V_c^{\mathcal{M}}(\pi_l) + (1 - \alpha_{l,1}) V_c^{\mathcal{M}}(\pi_s) \geq (1 - \alpha_{l,1}) \left( \frac{v_{l,s}}{2} - \varepsilon(L+2) \right) - \frac{\alpha_{l,1}}{1 - \gamma} \geq 0,$$

where the first inequality follows from the boundedness of $V_c^{\mathcal{M}}(\pi_l)$, and the second inequality follows by setting

$$\alpha_{l,1} = \frac{v_{l,s} - 2\varepsilon(L+2)}{v_{l,s} - 2\varepsilon(L+2) + 2/(1-\gamma)}.$$

**Inductive step.** By induction, $\pi_{l,m}$ is feasible for $m \geq 1$. Now, we will show that $\pi_{l,m+1}$ is also feasible. By Inequality (25), we have

$$\alpha_{l,m} V_c^{\mathcal{M}}(\pi) + (1 - \alpha_{l,m}) V_c^{\mathcal{M}}(\pi_s) \geq \frac{v_{l,m}}{2} - \varepsilon(L+2) \geq \frac{(1 - \alpha_{l,m}) v_{l,s}}{2} - \left( L + \frac{5}{2} \right) \varepsilon, \tag{27}$$

where the last inequality follows from $v_{l,m} = \alpha_{l,m} v_l + (1 - \alpha_{l,m}) v_{l,s}$ and the fact that $v_l \geq -\varepsilon$ by construction from Algorithm 1. Combining the above with Inequality (26), we further obtain

$$\alpha_{l,m} V_c^{\mathcal{M}}(\pi) - \alpha_{l,m} V_c^{\mathcal{M}}(\pi_s) \geq -\frac{(2 + \alpha_{l,m}) v_{l,s}}{2} - \left( 2L + \frac{9}{2} \right) \varepsilon.$$

Finally, the feasibility of the mixture policy $\pi_{l,m+1}$ follows from

$$\begin{aligned}
V_c^{\mathcal{M}}(\pi_{l,m+1}) &= \alpha_{l,m+1} V_c^{\mathcal{M}}(\pi_l) + (1 - \alpha_{l,m+1}) V_c^{\mathcal{M}}(\pi_s) \\
&= \alpha_{l,m} V_c^{\mathcal{M}}(\pi_l) + (1 - \alpha_{l,m}) V_c^{\mathcal{M}}(\pi_s) + (\alpha_{l,m+1} - \alpha_{l,m}) \left[ V_c^{\mathcal{M}}(\pi_l) - V_c^{\mathcal{M}}(\pi_s) \right] \\
&\geq \frac{(1 - \alpha_{l,m}) v_{l,s}}{2} - \left( L + \frac{5}{2} \right) \varepsilon - \left( \frac{\alpha_{l,m+1}}{\alpha_{l,m}} - 1 \right) \left( \frac{(2 + \alpha_{l,m}) v_{l,s}}{2} + \left( 2L + \frac{9}{2} \right) \varepsilon \right) \\
&\geq 0,
\end{aligned}$$

where the last inequality holds since

$$\frac{\alpha_{l,m+1}}{\alpha_{l,m}} = \frac{3 v_{l,s}}{(2 + \alpha_{l,m}) v_{l,s} + (4L+9)\varepsilon} \leq \frac{3 v_{l,s} + (2L+4)\varepsilon}{(2 + \alpha_{l,m}) v_{l,s} + (4L+9)\varepsilon}.$$

Hence, $\pi_{l,m+1}$ remains feasible in $\mathcal{M}$. By induction, all mixture policies $\pi_{l,m}$ generated during testing satisfy the constraint with probability at least $1 - \delta$. Thus, safe exploration is guaranteed throughout the testing phase with high probability. $\square$

## E. Proof of Theorem 4.2

*Proof of Theorem 4.2.* Lemma 4.5 shows that, with high probability, the following hold:

1. Safe exploration is ensured during testing.

2. At every iteration, the value estimate satisfies $u_l \geq V_r^{\mathcal{M}}(\pi^*) - 4\varepsilon L \left( 1 + \xi^{-1}(1-\gamma)^{-1} \right)$.

In the following analysis, we condition on these events. Since (i) guarantees safe exploration, we focus on bounding the reward regret.

Algorithm D considers a sequence of policies $\{\pi_l\}_{l=1}^E \subset \hat{\mathcal{U}}$, where $E \leq |\hat{\mathcal{U}}| \leq 2\mathcal{C}_\varepsilon(\mathcal{D}, \delta) \ln \frac{1}{\delta}$. For each $\pi_l$, the deployed policy $\pi_{l,m}$ is a mixture policy of $\pi_l$ and $\pi_s$ with mixture weight $\alpha_{l,m}$, where $m \in [0, m(l)]$. The value $m(l)$ is determined by the update rule in Algorithm 2, which satisfies

$$m(l) \leq \left\lceil \log_{\frac{2v_{l,s}+(4L+5)\varepsilon}{3v_{l,s}}} \varepsilon \right\rceil.$$

We denote by $\tau_{l,m}$ the first episode in which the mixture policy $\pi_{l,m}$ is used. For notational convenience, we also define $\tau_{l,\,m(l)+1} := \tau_{l+1,0}$, so that the iteration that stops using $\pi_l$ coincides with the starting iteration of the next policy $\pi_{l+1}$. We decompose the reward regret as

$$\text{Reg}_r^{\mathcal{M}}(K) = \sum_{l=1}^E \sum_{m=0}^{m(l)} \sum_{\tau=\tau_{l,m}}^{\tau_{l,m+1}-1} \left[ V_r^{\mathcal{M}}(\pi^*) - V_r^{\mathcal{M}}(\pi_{l,m}) \right]_+$$

$$= \sum_{l=1}^E \sum_{m=0}^{m(l)} \sum_{\tau=\tau_{l,m}}^{\tau_{l,m+1}-1} \Bigg( \underbrace{\left[ V_r^{\mathcal{M}}(\pi^*) - u_l \right]_+}_{(i)} + \underbrace{|u_l - (\alpha_{l,m}u_l + (1-\alpha_{l,m}u_{l,s}))|}_{(ii)} + \underbrace{|(\alpha_{l,m}u_l + (1-\alpha_{l,m}u_{l,s})) - R_\tau|}_{(iii)}$$

$$+ \underbrace{|R_\tau - V_r^{\mathcal{M}}(\pi_{l,m})|}_{(iv)} \Bigg).$$

**Term (i).** By Lemma 4.5,
$$\sum_{l=1}^E \sum_{m=0}^{m(l)} \sum_{\tau=\tau_{l,m}}^{\tau_{l,m+1}-1} (i) \leq 4\varepsilon L K \left( 1 + \frac{1}{\xi(1-\gamma)} \right).$$

**Calculation of weight $\alpha_{l,m}$.** From line 12 of Algorithm 2, we obtain the closed form of $\alpha_{l,m}$:

$$\alpha_{l,0} = 0, \qquad \alpha_{l,1} = \frac{v_{l,s} - 2\varepsilon(L+2)}{v_{l,s} - 2\varepsilon(L+2) + 2/(1-\gamma)},$$

$$\alpha_{l,m} = \frac{(v_{l,s} - (4L+9)\varepsilon)\alpha_{l,1}}{v_{l,s}\alpha_{l,1} + (v_{l,s} - (4L+9)\varepsilon - v_{l,s}\alpha_{l,1})C_l^m}, \qquad m \in [m(l)], \tag{28}$$

where

$$C_l := \frac{2v_{l,s} + (4L+9)\varepsilon}{3v_{l,s}} < 1$$

because $v_{l,s} \geq \xi > (4L+9)\varepsilon$ by assumption. Thus $\alpha_{l,m}$ is increasing in $m$ and upper bounded by $\lim_{m\to\infty} \alpha_{l,m} = 1 - \frac{(4L+9)\varepsilon}{v_{l,s}}$. Moreover,

$$v_{l,m} = \alpha_{l,m}v_l + (1-\alpha_{l,m})v_{l,s} \geq -\alpha_{l,m}\varepsilon + (1-\alpha_{l,m})v_{l,s} \geq \frac{(1-\alpha_{l,m})v_{l,s}}{2}, \tag{29}$$

where the last step uses the upper bound

$$\alpha_{l,m} \leq 1 - \frac{(4L+9)\varepsilon}{v_{l,s}} \leq \frac{v_{l,s}}{v_{l,s} + 2\varepsilon}.$$

When $m \geq \log_{C_l} \varepsilon$, we further obtain

$$\alpha_{l,m} \geq 1 - A_l\varepsilon, \text{ where } A_l := \frac{(4L+9-v_{l,s})\alpha_{l,1} + v_{l,s} - (4L+9)\varepsilon}{v_{l,s}\alpha_{l,1} + (v_{l,s} - (4L+9)\varepsilon - v_{l,s}\alpha_{l,1})\varepsilon}. \tag{30}$$

**Term (ii).** We have

$$(ii) = \sum_{l=1}^{E} \sum_{m=0}^{m(l)} \sum_{\tau=\tau_{l,m}}^{\tau_{l,m+1}-1} |u_l - (\alpha_{l,m} u_l + (1 - \alpha_{l,m}) u_{l,s})|$$

$$\leq \frac{2}{1-\gamma} \sum_{l=1}^{E} \sum_{m=0}^{m(l)} (\tau_{l,m+1} - \tau_{l,m})(1 - \alpha_{l,m})$$

$$\leq \frac{2}{1-\gamma} \sqrt{\sum_{l=1}^{E} \sum_{m=0}^{m(l)} (\tau_{l,m+1} - \tau_{l,m})} \sqrt{\sum_{l=1}^{E} \sum_{m=0}^{m(l)} (\tau_{l,m+1} - \tau_{l,m})(1 - \alpha_{l,m})^2}$$

$$= \frac{2\sqrt{K}}{1-\gamma} \sqrt{\sum_{l=1}^{E} \sum_{m=0}^{m(l)} (\tau_{l,m+1} - \tau_{l,m})(1 - \alpha_{l,m})^2},$$

where the first bound uses the fact that $v_l$ and $v_{l,s}$ are bounded, and the second uses the Cauchy-Schwarz inequality.

If $m(l) \leq \lfloor \log_{C_l} \varepsilon \rfloor$, then by line 11 of Algorithm 2 and Inequality (29),

$$\sum_{m=0}^{m(l)} (\tau_{l,m+1} - \tau_{l,m})(1 - \alpha_{l,m})^2 \leq \sum_{m=0}^{m(l)} \frac{32 \ln(4K/\delta)(1 - \alpha_{l,m})^2}{(1-\gamma)^2 v_{l,m}^2}$$

$$\leq \sum_{m=0}^{m(l)} \frac{128 \ln(4K/\delta)}{(1-\gamma)^2 v_{l,s}^2}$$

$$\leq \frac{128 \ln(4K/\delta) \log_{C_l} \varepsilon}{(1-\gamma)^2 v_{l,s}^2}. \tag{31}$$

If instead $m(l) = \lceil \log_{C_l} \varepsilon \rceil$, then applying Inequalities (30) and (31) gives

$$\sum_{m=0}^{\lceil \log_{C_l} \varepsilon \rceil} (\tau_{l,m+1} - \tau_{l,m})(1 - \alpha_{l,m})^2$$

$$\leq \sum_{m=0}^{\lfloor \log_{C_l} \varepsilon \rfloor} (\tau_{l,m+1} - \tau_{l,m})(1 - \alpha_{l,m})^2 + (\tau_{l,\lceil \log_{C_l} \varepsilon \rceil + 1} - \tau_{l,\lceil \log_{C_l} \varepsilon \rceil})(1 - \alpha_{l,\lceil \log_{C_l} \varepsilon \rceil})^2$$

$$\leq \frac{128 \ln(4K/\delta) \log_{C_l} \varepsilon}{(1-\gamma)^2 v_{l,s}^2} + (\tau_{l,\lceil \log_{C_l} \varepsilon \rceil + 1} - \tau_{l,\lceil \log_{C_l} \varepsilon \rceil}) A_l^2 \varepsilon^2. \tag{32}$$

Combining Inequalities (31) and (32), we obtain

$$(ii) \leq \frac{2\sqrt{K}}{1-\gamma} \sqrt{\sum_{l=1}^{E} \frac{128 \ln(4K/\delta) \log_{C_l} \varepsilon}{(1-\gamma) v_{l,s}^2} + (\tau_{l,\lceil \log_{C_l} \varepsilon \rceil + 1} - \tau_{l,\lceil \log_{C_l} \varepsilon \rceil}) A_l^2 \varepsilon^2}$$

$$\leq \frac{2\sqrt{K}}{1-\gamma} \left( \sqrt{\sum_{l=1}^{E} \frac{128 \ln(4K/\delta) \log_{C_l} \varepsilon}{(1-\gamma)^2 v_{l,s}^2}} + \sqrt{\sum_{l=1}^{E} (\tau_{l,\lceil \log_{C_l} \varepsilon \rceil + 1} - \tau_{l,\lceil \log_{C_l} \varepsilon \rceil}) A_l^2 \varepsilon^2} \right)$$

$$\leq \frac{\max_l 2K A_l \varepsilon}{1-\gamma} + \frac{\max_l 32 \sqrt{EK \ln(4K/\delta) \log_{C_l} \varepsilon}}{(1-\gamma)^2 \min_l v_{l,s}}.$$

**Term (iii).** From the elimination condition (line 7 of Algorithm 2),

$$(iii) \leq \sum_{l=1}^{E} \sum_{m=0}^{m(l)} \sqrt{\frac{2(\tau_{m+1} - \tau_m) \ln(4K/\delta)}{(1-\gamma)^2}} + \varepsilon(L+1)(\tau_{m+1} - \tau_m)$$

$$\leq \varepsilon(L+1)K + \max_l \sqrt{\frac{2EK\ln(4K/\delta)\lceil\log_{C_l}\varepsilon\rceil}{(1-\gamma)^2}},$$

where the last inequality uses the Cauchy–Schwarz inequality and $m(l) \leq \lceil\log_{C_l}\varepsilon\rceil$.

**Term (iv).** By (Ni & Kamgarpour, 2025, Proposition 3.3), and by further setting the truncated horizon $H = \mathcal{O}\left(\frac{\ln\varepsilon^{-1}}{1-\gamma}\right)$, we have

$$(iv) \leq \sum_{l=1}^{E}\sum_{m=0}^{m(l)}\sqrt{\frac{2(\tau_{l,m+1}-\tau_{l,m})\ln(4K/\delta)}{(1-\gamma)^2}} + \varepsilon(\tau_{l,m+1}-\tau_{l,m})$$

$$\leq \varepsilon K + \max_l \sqrt{\frac{2EK\ln(4K/\delta)\lceil\log_{C_l}\varepsilon\rceil}{(1-\gamma)^2}},$$

where the last inequality uses the Cauchy–Schwarz inequality and $m(l) \leq \lceil\log_{C_l}\varepsilon\rceil$.

**Final bound.** Combining terms (i)–(iv) gives

$$\text{Reg}_r^{\mathcal{M}}(K) \leq \mathcal{O}\left(\frac{\varepsilon LK}{\xi(1-\gamma)} + \frac{\max_l A_l K\varepsilon}{(1-\gamma)} + \frac{\max_l \sqrt{C_\varepsilon(\mathcal{D},\delta)K\ln\frac{K}{\delta}\ln\frac{1}{\delta}\log_{C_l}\varepsilon}}{\min_l(1-\gamma)^2 v_{l,s}}\right)$$

$$\leq \mathcal{O}\left(\frac{\varepsilon LK}{\xi(1-\gamma)} + \frac{\sqrt{C_\varepsilon(\mathcal{D},\delta)K\ln\frac{K}{\delta}\ln\frac{1}{\delta}\ln\frac{1}{\varepsilon}}}{(1-\gamma)^2\xi}\right),$$

where the first inequality follows from $E \leq 2\mathcal{C}_\varepsilon(\mathcal{D},\delta)\ln\frac{1}{\delta}$, and the second inequality follows from $\min_l v_{l,s} \geq \xi$,

$$A_l = \mathcal{O}\left(\frac{L}{v_{l,s}} + \frac{1}{\alpha_{l,s}}\right) = \mathcal{O}\left(\frac{L}{\xi}\right),$$

and

$$\log_{C_l}\varepsilon = \log_{\frac{3v_{l,s}}{2v_{l,s}+(4L+9)\varepsilon}}\frac{1}{\varepsilon} = \mathcal{O}\left(\log_{\frac{3\xi}{2\xi+(4L+9)\varepsilon}}\frac{1}{\varepsilon}\right) = \mathcal{O}\left(\ln\frac{1}{\varepsilon}\right)$$

because of $v_{l,s} \geq \xi \geq 2(4L+9)\varepsilon$. Since $L = \frac{1}{1-\gamma} + \frac{2\gamma}{(1-\gamma)^2}$, we conclude that

$$\text{Reg}_r^{\mathcal{M}}(K) \leq \tilde{\mathcal{O}}\left(\frac{\sqrt{\mathcal{C}_\varepsilon(\mathcal{D},\delta)K}}{(1-\gamma)^2\xi} + \frac{\varepsilon K}{\xi(1-\gamma)^3}\right).$$

$\square$

# F. Constrained Meta RL without Safe Exploration: An Extension of (Ye et al., 2023)

Ye et al. introduced the PCE algorithm in the meta RL setting. With a simple modification that accounts for constraint values in addition to rewards during both the training and testing phases, their algorithm can be extended to the constrained meta RL setting without safe exploration guarantees. In this section, we detail this extension and provide guarantees that it achieves sublinear test-time regret for both reward and constraints. The reward regret is defined in (1), while the constraint regret is defined as

$$\text{Reg}_c^{\mathcal{M}_{\text{test}}}(K) := \sum_{k=1}^{K}\left[V_c^{\mathcal{M}_{\text{test}}}(\pi_k)\right]_-,$$

where $\{\pi_k\}_{k=1}^{K}$ denotes the sequence of policies generated during the testing phase, and $\mathcal{M}_{\text{test}}$ denotes the test CMDP.

**Training phase**   Our training-phase Algorithm 1 first constructs the CMDP set $\mathcal{U}$ and then learns both the optimal and feasible policies, including the feasible policy $\pi_s$. Since safe exploration is not required during training, we ignore entries related to $\pi_s$, and denote the resulting set as

$$\hat{\mathcal{U}} := \{(\pi, V_r^{\mathcal{M}}(\pi), V_c^{\mathcal{M}}(\pi)) \mid \mathcal{M} \in \mathcal{U}\},$$

where all entries are obtained via the oracle $\mathbb{O}_l$, and the guarantees for this set are provided in Lemma 3.5. Although we believe that the training algorithm of Ye et al. could, in principle, be adapted to directly learn a set of optimal and feasible policies without first constructing a CMDP set, we do not provide a formal proof for this modification. Such a proof would likely follow arguments similar to those in Lemma 3.5. Nonetheless, we expect that this direct adaptation would yield similar guarantees for the resulting policy set.

**Testing phase**   If we set $\alpha_{l,m} = 1$ in our Algorithm 2, the mixture policy $\pi_{l,m}$ reduces to $\pi_l$. In this case, our method directly extends the test-phase algorithm of (Ye et al., 2023), with the addition of a constraint check:

$$\left|\frac{\sum_{j=k_0}^k C_j}{k - k_0 + 1} - v_l\right| \geq \sqrt{\frac{2\ln(4K/\delta)}{(k-k_0+1)(1-\gamma)^2}} + \varepsilon(L+1).$$

This adapted version is provided in Algorithm 4. However, safe exploration is not guaranteed, since any policy $\pi_l \in \hat{\mathcal{U}}$ may be infeasible during testing.

---

**Algorithm 4** Testing phase adapted from PCE (Ye et al., 2023)

---

1: **Input:** Policy–value set $\hat{\mathcal{U}}$ and from Alg. 1, number of iterations $K$, truncated horizon $H$, confidence level $\delta \in (0,1)$.
2: **Initialize:** Set counters $l = 1$ and $k_0 = 1$.
3: **for** $k = 1, \ldots, K$ **do**
4:   Select policy with the highest reward value: $(\pi_l, u_l, v_l) = \arg\max_{(\pi,u,v)\in\hat{\mathcal{U}}_l} u$, and set $\pi_k = \pi_l$.
5:   Sample a trajectory $\tau_H$ using $\pi_l$ and compute cumulative reward value $R_k$ and constraint value $C_k$.
6:   **if** $\left|\frac{\sum_{j=k_0}^k R_j}{k-k_0+1} - u_l\right| \geq \sqrt{\frac{2\ln(4K/\delta)}{(k-k_0+1)(1-\gamma)^2}} + \varepsilon(L+1)$ **or** $\left|\frac{\sum_{j=k_0}^k C_j}{k-k_0+1} - v_l\right| \geq \sqrt{\frac{2\ln(4K/\delta)}{(k-k_0+1)(1-\gamma)^2}} + \varepsilon(L+1)$ **then**
7:     Update policy-value set: $\hat{\mathcal{U}} = \hat{\mathcal{U}} \setminus \{(\pi_l, u_l, v_l)\}$
8:     Update counters: $l = l + 1$, $k_0 = k + 1$
9:   **end if**
10: **end for**
11: **Return:** Final policy $\pi_{out} = \frac{1}{K}\sum_{k=1}^K \pi_k$.

---

Next, we proceed to prove the regret guarantees for Algorithm 4, as stated in Corollary F.1.

**Corollary F.1.** *Let Assumption 3.3 hold and set $(8L+18)\varepsilon \leq \xi$ and the truncated horizon as $H = \tilde{\mathcal{O}}((1-\gamma)^{-1})$. Then, for any test CMDP $\mathcal{M} \sim \mathcal{D}$, Algorithms 1 and 4 satisfies the following with probability at least $1 - 26\delta - 9\delta\ln C_\varepsilon(\mathcal{D}, \delta)$,*

$$\sum_{k=1}^k [V_r^{\mathcal{M}}(\pi^*) - V_r^{\mathcal{M}}(\pi_k)]_+ \leq \tilde{\mathcal{O}}\left(\frac{\sqrt{\mathcal{C}_\varepsilon(\mathcal{D}, \delta)K}}{(1-\gamma)} + \frac{\varepsilon K}{\xi(1-\gamma)^3}\right),$$

$$\sum_{k=1}^k [V_c^{\mathcal{M}}(\pi_k)]_- \leq \tilde{\mathcal{O}}\left(\frac{\sqrt{\mathcal{C}_\varepsilon(\mathcal{D}, \delta)K}}{(1-\gamma)} + \frac{\varepsilon K}{(1-\gamma)^2}\right).$$

*Proof of Corollary F.1.* In Lemma 3.5, we show that for any $\mathcal{M} \sim \mathcal{D}$, with probability at least $1 - 25\delta - 9\delta\ln C_\varepsilon(\mathcal{D}, \delta)$, there exists a tuple $(\pi, u, v) \in \hat{\Pi}$ such that

$$\max\left\{\left|V_r^{\mathcal{M}}(\pi) - u\right|, \left|V_c^{\mathcal{M}}(\pi) - v\right|\right\} \leq \varepsilon L, \tag{33}$$

$$\left|u - V_r^{\mathcal{M}}(\pi^*)\right| \leq 4\varepsilon L\left(1 + \frac{1}{\xi(1-\gamma)}\right). \tag{34}$$

In the following analysis, we focus on the case where the events of Lemma 3.5 hold. Algorithm 4 considers a sequence of policies $\{\pi_l\}_{l=1}^E \subset \hat{\mathcal{U}}$, where $E \leq |\hat{\mathcal{U}}| \leq 2\mathcal{C}_\varepsilon(\mathcal{D}, \delta) \ln \frac{1}{\delta}$. We denote by $\tau_l$ the first episode in which policy $\pi_l$ is used. Next, we show that for a sampled CMDP $\mathcal{M}$, with probability at least $1 - \delta$, at every iteration of Algorithm 4, the value estimate satisfies

$$v_l \geq V_r^{\mathcal{M}}(\pi^*) - 4\varepsilon L \left(1 + \frac{1}{\xi(1-\gamma)}\right).$$

Similar to Step 2 in the proof of Lemma 4.5, by applying concentration bounds (Ni & Kamgarpour, 2025, Proposition 3.3), we obtain that with probability at least $1 - \delta$,

$$\left| \frac{1}{k - k_0 + 1} \sum_{\tau=k_0}^{k} R_\tau - u_l \right| \leq \sqrt{\frac{2\ln(4K/\delta)}{(k - k_0 + 1)(1-\gamma)^2}} + \varepsilon(L + 1),$$

$$\left| \frac{1}{k - k_0 + 1} \sum_{\tau=k_0}^{k} C_\tau - v_l \right| \leq \sqrt{\frac{2\ln(4K/\delta)}{(k - k_0 + 1)(1-\gamma)^2}} + \varepsilon(L + 1).$$

By the elimination rule (line 6 of Algorithm 4), the above inequalities guarantee that $(\pi, u, v)$ will *not* be eliminated from $\hat{\mathcal{U}}$ with probability at least $1 - \delta$. Combining Inequality (34) with the fact that in each iteration of Algorithm 4, $u_l = \max_{(\pi,u,v)\in\hat{\mathcal{U}}_l} u$, we obtain

$$u_l \geq V_r^{\mathcal{M}}(\pi^*) - 4\varepsilon L \left(1 + \frac{1}{\xi(1-\gamma)}\right). \tag{35}$$

Now we are ready to prove Corollary F.1. We start by decomposing the reward regret:

$$
\begin{aligned}
\sum_{k=1}^{K} [V_r^{\mathcal{M}}(\pi^*) - V_r^{\mathcal{M}}(\pi_k)]_+ &= \sum_{l=1}^{E} \sum_{\tau=\tau_l}^{\tau_{l+1}-1} \left[ V_r^{\mathcal{M}}(\pi^*) - u_l + u_l - R_\tau + R_\tau - V_r^{\mathcal{M}}(\pi_l) \right]_+ \\
&\leq \sum_{l=1}^{E} \sum_{\tau=\tau_l}^{\tau_{l+1}-1} \left[ V_r^{\mathcal{M}}(\pi^*) - u_l \right]_+ + \sum_{l=1}^{E} \left| \sum_{\tau=\tau_l}^{\tau_{l+1}-1} (u_l - R_\tau) \right| + \sum_{l=1}^{E} \left| \sum_{\tau=\tau_l}^{\tau_{l+1}-1} (R_\tau - V_r^{\mathcal{M}}(\pi_l)) \right| \\
&\leq 4\varepsilon K L \left(1 + \frac{1}{\xi(1-\gamma)}\right) + \underbrace{\sum_{l=1}^{E} \left| \sum_{\tau=\tau_l}^{\tau_{l+1}-1} (u_l - R_\tau) \right|}_{(i)_r} + \underbrace{\sum_{l=1}^{E} \left| \sum_{\tau=\tau_l}^{\tau_{l+1}-1} (R_\tau - V_r^{\mathcal{M}}(\pi_l)) \right|}_{(ii)_r},
\end{aligned}
$$

where the last inequality uses Inequality (35).

For the constraint regret, we have

$$
\begin{aligned}
\sum_{k=1}^{K} \left[ V_c^{\mathcal{M}}(\pi_k) \right]_- &= \sum_{k=1}^{K} \left[ -V_c^{\mathcal{M}}(\pi_k) \right]_+ \\
&= \sum_{l=1}^{E} \sum_{\tau=\tau_l}^{\tau_{l+1}-1} \left[ -V_c^{\mathcal{M}}(\pi_l) + C_\tau - C_\tau + v_l - v_l \right]_+ \\
&\leq \sum_{l=1}^{E} \left| \sum_{\tau=\tau_l}^{\tau_{l+1}-1} (v_l - C_\tau) \right| + \sum_{l=1}^{E} \left| \sum_{\tau=\tau_l}^{\tau_{l+1}-1} (C_\tau - V_c^{\mathcal{M}}(\pi_l)) \right| + \sum_{l=1}^{E} \sum_{\tau=\tau_l}^{\tau_{l+1}-1} [-v_l]_+ \\
&\leq \underbrace{\sum_{l=1}^{E} \left| \sum_{\tau=\tau_l}^{\tau_{l+1}-1} (v_l - C_\tau) \right|}_{(i)_c} + \underbrace{\sum_{l=1}^{E} \left| \sum_{\tau=\tau_l}^{\tau_{l+1}-1} (C_\tau - V_c^{\mathcal{M}}(\pi_l)) \right|}_{(ii)_c} + K\varepsilon,
\end{aligned}
$$

where the last inequality follows from $v_l \geq -\varepsilon$ by construction (Algorithm 1).

For terms $(i)_r$ and $(i)_c$, by the elimination condition (line 6 of Algorithm 4), we have

$$\max\{(i)_r, (i)_c\} \le \sum_{l=1}^{E} \sqrt{\frac{2(\tau_{l+1} - \tau_l)\ln(4K/\delta)}{(1-\gamma)^2}} + \varepsilon(L+1)K.$$

For terms $(ii)_r$ and $(ii)_c$, using the concentration bound in (Ni & Kamgarpour, 2025, Proposition 3.3), we obtain

$$\max\{(ii)_r, (ii)_c\} \le \sum_{l=1}^{E} \sqrt{\frac{2(\tau_{l+1} - \tau_l)\ln(4K/\delta)}{(1-\gamma)^2}} + \varepsilon K.$$

Combining terms $(i)_r$ and $(ii)_r$ gives

$$\sum_{k=1}^{K}[V_r^{\mathcal{M}}(\pi^*) - V_r^{\mathcal{M}}(\pi_k)]_+ \le \varepsilon K\left(2 + 5L + \frac{4L}{\xi(1-\gamma)}\right) + 2\sum_{l=1}^{E}\sqrt{\frac{2(\tau_{l+1} - \tau_l)\ln(4K/\delta)}{(1-\gamma)^2}}$$

$$\overset{(1)}{\le} \varepsilon K\left(2 + 5L + \frac{4L}{\xi(1-\gamma)}\right) + 2\sqrt{\frac{2KE\ln(4K/\delta)}{(1-\gamma)^2}}$$

$$\overset{(2)}{\le} \tilde{\mathcal{O}}\left(\frac{\sqrt{\mathcal{C}_\varepsilon(\mathcal{D}, \delta)K}}{1-\gamma} + \frac{\varepsilon K}{\xi(1-\gamma)^3}\right),$$

where (1) uses the Cauchy–Schwarz inequality, and (2) uses $E \le |\hat{\mathcal{U}}| \le 2\mathcal{C}_\varepsilon(\mathcal{D}, \delta)\ln\frac{1}{\delta}$ and $L = \frac{1}{1-\gamma} + \frac{2\gamma}{(1-\gamma)^2}$.

Similarly, combining terms $(i)_c$ and $(ii)_c$ gives

$$\sum_{k=1}^{K}\left[V_c^{\mathcal{M}}(\pi_k)\right]_- \le \varepsilon K(3 + L) + 2\sum_{l=1}^{E}\sqrt{\frac{2(\tau_{l+1} - \tau_l)\ln(4K/\delta)}{(1-\gamma)^2}}$$

$$\le \varepsilon K(3 + L) + 2\sqrt{\frac{2KE\ln(4K/\delta)}{(1-\gamma)^2}}$$

$$\le \tilde{\mathcal{O}}\left(\frac{\sqrt{\mathcal{C}_\varepsilon(\mathcal{D}, \delta)K}}{1-\gamma} + \frac{\varepsilon K}{(1-\gamma)^2}\right).$$

$\square$

## G. Proof of Theorem 5.1

In the following, we present a more explicit version of Theorem 5.1, which specifies how the constants $\gamma_0$, $\varepsilon_0$, and $\delta_0$ depend on the problem size.

**Theorem G.1.** *There exists a distribution $\mathcal{D}$ over the CMDP class $\mathcal{M}_{all}$ and constants $\gamma_0 \in \left(1 - (\log|\mathcal{S}|)^{-1}, 1\right)$, $0 \le \varepsilon_0 \le \min\left\{(1-\gamma)^{-1}, \gamma\xi^{-1}(1-\gamma)^{-2}\right\}$, and $\delta_0 \in (0, 1)$ such that for any $\gamma \in (\gamma_0, 1)$, $\varepsilon \in (0, \varepsilon_0)$, $\delta \in (0, \delta_0)$, any algorithm requires at least $\tilde{\Omega}\left(\xi^{-2}\varepsilon^{-2}(1-\gamma)^{-5}\mathcal{C}_{\xi\varepsilon(1-\gamma)^3}(\mathcal{D}, \delta)\right)$ samples from the generative model such that for any $\mathcal{M} \sim \mathcal{D}$, the output policy is $\varepsilon$-optimal and feasible with high probability.*

*Proof.* To establish a lower bound on the sample complexity over the CMDP set $\Omega$, we first construct a hard instance $\mathcal{M}_0$. Next, we construct alternative CMDPs $\mathcal{M}_i$ that differ from $\mathcal{M}_0$ in specific transition probabilities and define a uniform distribution over this set. Using this construction, we then derive a lower bound on the number of samples any algorithm must collect to identify an $\varepsilon$-optimal and feasible policy for a hard instance with high probability.

**Construction of the hard instance:** The hard instance $\mathcal{M}_0$ was initially proposed in (Feng et al., 2019) to study lower bounds on sample complexity in transfer RL. It was later adapted by (Vaswani et al., 2022) to the constrained setting by incorporating CMDPs, where the lower bound must account for both reward maximization and constraint satisfaction. In this work, we further construct $\mathcal{M}_0$ following the CMDP design in (Vaswani et al., 2022, Theorem 8), extending it to the constrained meta RL setting with several parameters adapted accordingly. Specifically, we scale the state and action space with the size of the task family $\mathcal{M}_{\text{all}}$, where each CMDP in $\mathcal{M}_{\text{all}}$ differs in its transition dynamics. Since the agent must identify each CMDP and learn its optimal feasible policy, the resulting sample complexity lower bound grows with the size of the task family $\mathcal{M}_{\text{all}}$. Without loss of generality, we assume $|\Omega| = 2^m$ for some integer $m > 0$. We then construct a CMDP $\mathcal{M}_0$ with a total of $3 \cdot 2^{m+1} - 1$ states, as illustrated in Figure 3.

**Structure.** Let $o_0$ be the fixed initial state, where $\rho(o_0) = 1$. From $o_0$, there is a deterministic path of length $m+1$ leading to each gate state $s_i$ for $i \in \{0, 1, \ldots, |\Omega|-1\}$. All transitions are deterministic except at states $\tilde{s}_i$. For $i \in \{0, 1, \ldots, |\Omega|-1\}$, taking action $a_0$ at $\tilde{s}_i$ leads back to $\tilde{s}_i$ with probability $p_i$ and to state $z_i$ with probability $1 - p_i$. We also include $2|\Omega| - 1$ routing states $o_0, o_1, \ldots$ forming a binary tree: in each routing state $o_i$ there are two actions $a_0, a_1$, and

$$(o_i, a_j) \to o_{2i+j+1}, \qquad i < |\Omega| - 1.$$

Each state $o_{|\Omega|+k-1}$ transitions deterministically to $s_k$ for $k = 0, \ldots, |\Omega| - 1$. Reward and constraint values are zero in all routing states. Because each $s_k$ has a unique deterministic path of length $m+1$ from $o_0$, we require $\gamma \geq 1 - 1/(cm)$ for some constant $c$, implying

$$\gamma^{\Theta(m)} = \Theta(1).$$

For the reward and constraint values at the states $s_i, \tilde{s}_i, z_i, z_i'$, we mark them in green and red, respectively, in Figure 3.

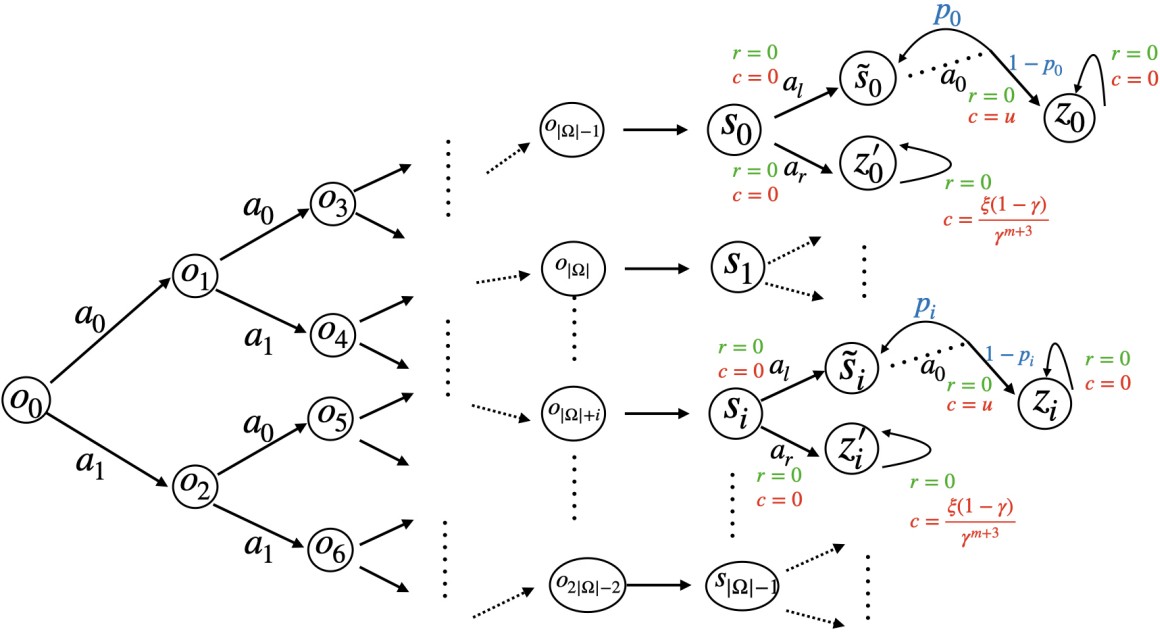

Figure 3: The hard CMDP instance $\mathcal{M}_0$ from (Vaswani et al., 2022), adapted to our constrained meta RL setting.

**Alternative CMDPs.** For each $i \in [|\Omega| - 1]$, we construct $\mathcal{M}_i$ to be identical to $\mathcal{M}_0$ except for the transition parameter $p_i$. Formally:

- *Null case $\mathcal{M}_0$*: $p_0 = q_1$ and $p_i = q_0$ for all $i \geq 1$.

- *Alternative case $\mathcal{M}_i$ where $i \in [|\Omega| - 1]$: $p_0 = q_1$, $p_j = q_0$ for all $j \neq i$, and $p_i = q_2$.*

The parameters $q_0, q_1, q_2$ are defined as

$$q_0 = \frac{1 - c_1(1 - \gamma)}{\gamma}, \qquad q_1 = q_0 + \alpha_1, \qquad q_2 = q_0 + \alpha_2,$$

where

$$\alpha_1 = \frac{c_2(1 - \gamma q_0)^2 \epsilon'}{\gamma}, \qquad \alpha_2 = \frac{c_3(1 - \gamma q_0)^2 \epsilon'}{\gamma}, \qquad \epsilon' = (1 - \gamma)\zeta\epsilon,$$

for some absolute constants $c_1, c_2, c_3 > 0$, chosen such that

$$\alpha_1, \alpha_2 \in (0, \min\{q_0, 1 - q_0\}/2).$$

These choices guarantee

$$\frac{1}{1 - \gamma q_0} < \frac{1}{1 - \gamma q_1} < \frac{1}{1 - \gamma q_2} = \frac{1}{1 - \gamma q_0} + c_4 \epsilon'.$$

for some constant $c_4 > 0$, and

$$\left| \frac{1}{1 - \gamma q_1} - \frac{1}{1 - \gamma q_0} \right| = \Theta(\epsilon'), \qquad \left| \frac{1}{1 - \gamma q_1} - \frac{1}{1 - \gamma q_2} \right| = \Theta(\epsilon').$$

We also set the constraint value $u$ in Figure 3 as

$$u = \frac{-(1 - \gamma q_0)x}{\gamma^{m+3}}, \quad \text{where } x = \Theta(\xi).$$

**Distribution.** Let $\mathcal{D}$ denote the uniform distribution over the CMDP set $\Omega$. The pairwise distance between any two CMDPs in $\Omega$ can be computed as

$$d(\mathcal{M}_i, \mathcal{M}_j) = \alpha_2.$$

When the discount factor $\gamma$ is sufficiently close to 1, we have

$$\xi\varepsilon(1 - \gamma)^3 \leq \alpha_2 = \frac{c_3\,\xi(1 - \gamma)\varepsilon(1 - \gamma q_0)^2}{\gamma}.$$

Since the radius $\xi\varepsilon(1 - \gamma)^3$ is smaller than the distance between any two CMDPs in $\mathcal{M}_{\text{all}}$, we have

$$\mathcal{C}_{\xi\varepsilon(1-\gamma)^3}(\mathcal{D}, \delta) = |\Omega|. \tag{36}$$

**Sample complexity lower bound.** Let $N_{i,a_0}$ denote the number of times algorithm $\mathbb{A}$ samples $(s_i, a_0)$. By (Vaswani et al., 2022, Lemma 20), if

$$\mathbb{P}\big[\mathbb{A} \text{ returns an } \varepsilon\text{-optimal and feasible policy in } \mathcal{M}_i \mid \mathcal{M}_i\big] \geq 1 - \delta',$$

then

$$\mathbb{E}_{\mathcal{M}_0}[N_{i,a_0}] \geq \frac{c_{10}\log(1/\delta')}{(1 - \gamma)^3\epsilon'^2}, \tag{37}$$

for an absolute constant $c_{10} > 0$. If instead

$$\mathbb{P}_{\mathcal{M}_i \sim \mathcal{D}}\big[\mathbb{A} \text{ returns an } \varepsilon\text{-optimal and feasible policy in } \mathcal{M}_i\big] \geq 1 - \delta,$$

then, since $\mathcal{D}$ is the uniform distribution over $\Omega$, we have

$$\min_i \mathbb{P}\big[\mathbb{A} \text{ returns an } \varepsilon\text{-optimal and feasible policy in } \mathcal{M}_i \mid \mathcal{M}_i\big] \geq 1 - |\Omega|\,\delta.$$

Combining this with Inequality (37) and linearity of expectation yields:

$$\mathbb{E}_{\mathcal{M}_0}\Big[\sum_i N_{i,a_0}\Big] \geq \frac{c_{10}(|\Omega|-1)\big(\log(1/\delta)+\log(1/|\Omega|)\big)}{(1-\gamma)^3\epsilon'^2}$$

$$\overset{(i)}{=} \Omega\left(\frac{\mathcal{C}_{\frac{\xi\varepsilon(1-\gamma)^3}{2}}(\mathcal{D},\delta)(\log\delta^{-1}+\log\mathcal{C}_{\xi\varepsilon(1-\gamma)^3}(\mathcal{D},\delta)^{-1})}{\xi^2\varepsilon^2(1-\gamma)^5}\right)$$

$$= \tilde{\Omega}\left(\frac{\mathcal{C}_{\xi\varepsilon(1-\gamma)^3}(\mathcal{D},\delta)}{\xi^2\varepsilon^2(1-\gamma)^5}\right),$$

where step $(i)$ uses Equation (36). $\qquad\square$

## H. Implementation Details

In this section, we provide implementation details for the training algorithm. All experiments were conducted on a MacBook Pro equipped with an Apple M1 Pro chip and 32 GB of RAM.

### H.1. Oracle implementations for gridworld experiments

Here, we describe how the oracles required by Algorithm 1 are implemented in the gridworld environment. Each CMDP $\mathcal{M} \in \mathcal{M}_{\text{all}}$ differs only in its noise level. Accordingly, the CMDP check oracle (Definition 3.1) is implemented by directly comparing the noise levels of the corresponding CMDPs. For the optimal policy oracle (Definition 3.2), we solve a linear program to compute the optimal policy. This is feasible because the model is fully known during training and the state–action space is small, allowing efficient computation of the optimal solution. Finally, for the simultaneously feasible policy oracle (Definition 3.4), we leverage the monotonicity of feasibility with respect to the noise level: any policy that is feasible for the CMDP with the highest noise level ($i = 0.5$) is also feasible for all CMDPs with $i \in [0, 0.5]$. Therefore, we implement the simultaneously feasible policy oracle by computing the optimal policy for $\mathcal{M}_{0.5}$.

### H.2. Additional experiments on Gym

In this section, we provide additional empirical validation in the Gym library on high-dimensional locomotion tasks, namely Hopper and Half-Cheetah. The environments are described as follows:

1. **Hopper.** The Hopper task has a 12-dimensional state space and a 3-dimensional action space. The reward is defined as the negative absolute difference between the agent's velocity and a task-specific target velocity. The task distribution over target velocities is a truncated distribution on $[0, 1]$ with mean 0.5 and variance 0.1. The cost corresponds to the control effort of the robot.

2. **Half-Cheetah.** The Half-Cheetah task has a 17-dimensional state space and a 6-dimensional action space. The reward is defined as the negative absolute difference between the agent's velocity and a task-specific target velocity. The task distribution is a truncated distribution on $[0, 2]$ with mean 1 and variance 0.3. The cost is defined via a safety constraint $h_{\text{cheetah}} - h_0 \leq d_\tau$, i.e., a violation occurs when the agent's head height exceeds a threshold $h_0$.

We build on the implementation of (Xu & Zhu, 2026) and compare against four baselines across 20 test tasks per environment, each drawn from the corresponding task distribution: (a) MAML (Finn et al., 2017) with a constraint penalty, (b) meta-CPO (Cho & Sun, 2024), (c) SafeMeta (Xu & Zhu, 2026), and (d) CPO (Achiam et al., 2017). Methods (a,b,c) are constrained meta-RL approaches, while (d) is a standard online RL algorithm.

As shown in Figure 4, our method is the only approach that simultaneously improves reward while ensuring safe exploration. In contrast, existing methods either violate safety (e.g., MAML with constraints) or fail to consistently improve performance in more challenging environments (e.g., Half-Cheetah).

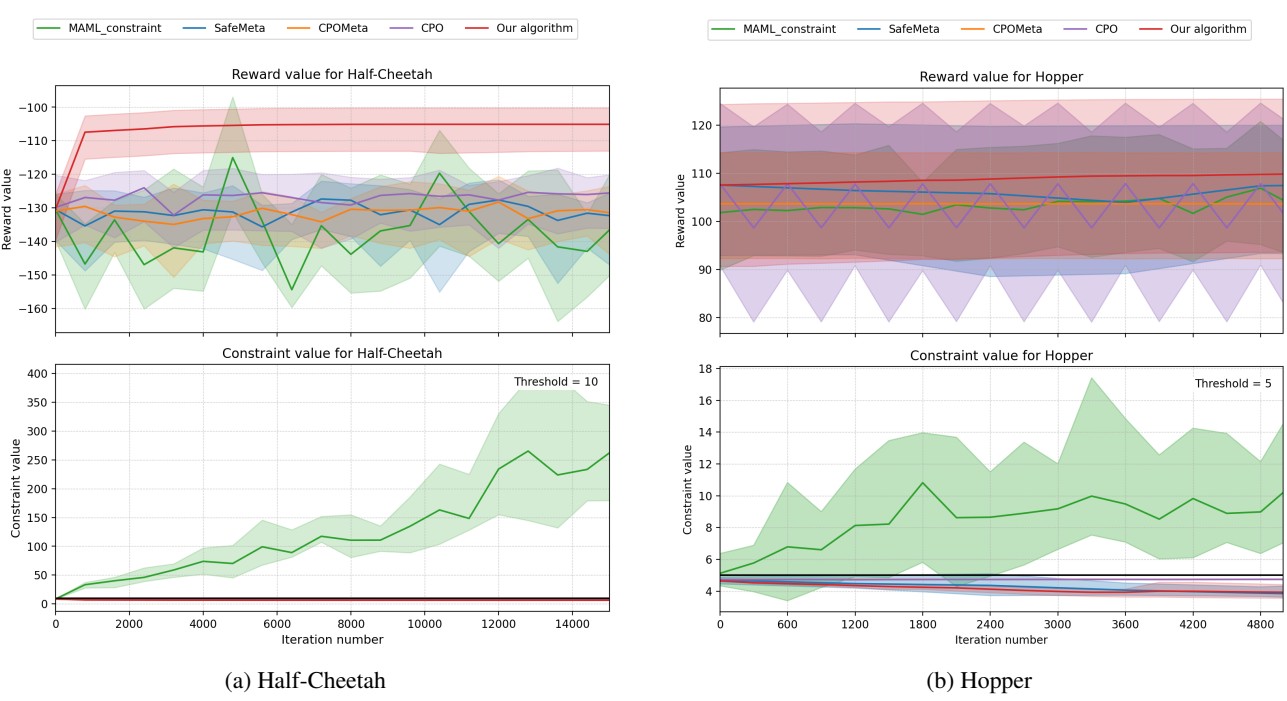

(a) Half-Cheetah                                        (b) Hopper

Figure 4: Comparison of Half-Cheetah and Hopper environments

