# OpenReview forum: "Constrained Meta Reinforcement Learning with Provable Test-Time Safety"
_ICML.cc/2026/Conference — ICML 2026 regular_

### Official Review · Reviewer_65Bn · 2026-02-16

**Soundness:** 3
**Presentation:** 3
**Significance:** 1
**Originality:** 2
**Overall Recommendation:** 4
**Confidence:** 3

**Summary:**

This paper addresses a constrained meta reinforcement learning problem, where an agent needs to learn optimal policies while ensuring
feasibility throughout the testing process. The authors propose an algorithm that refines policies learned during
training that enjoy provable safe exploration and sample complexity guarantees.

**Compliance With Llm Reviewing Policy:**

Affirmed.

**Final Justification:**

I feel it is very difficult to evaluate this paper. While I found the theoretical analyses of this paper useful, the empirical experiments are rather limited. I consider that every theoretical paper does not require a comprehensive experiments. However, since the most related work has conducted experiments in large-scale environment, I feel this paper can do the similar level of experiments.

**Key Questions For Authors:**

1. Could you tell me the reason why the authors use grid-world in your empirical evaluation? Are there any reasons why typical safe RL benchmark (e.g., Safety-Gym) cannot be used as benchmark tasks?

1. Regarding Table 1, how should we interpret the dependence on $C_\varepsilon$? Could you elaborate how large $C_\varepsilon$ is?

**Limitations:**

I think the authors have not discussed the limitations.

**Strengths And Weaknesses:**

## Strengths

1. This paper is well-written and easy to follow.
1. The theoretical results and their proofs are rigorously presented with solid mathematics.
1. The assumptions behind theoretical results are mild and I feel all of them are likely to hold in practice.
1. While theorems look somewhat incremental from those in typical safe RL literature, I feel Theorems 4.2 and 5.1 are indeed useful and new.

## Weaknesses

1. The empirical evaluation has been conducted in only grid-world environment. I think that the proposed method can be easily applied to more complicated tasks, and such an extended experiments should strengthen the contribution of this paper.
1. Since safe MetaRL (Xu and Zhu, 2025) conducted experiments on Safety-Gymnasium, so I consider that the proposed method in this paper should be evaluated on such a continuous control tasks.

---

> ### Author Rebuttal · Authors · 2026-03-31
>
> We sincerely thank the reviewer for the positive assessment and for recognizing the rigor of our theoretical results. We address your questions below.
>
> 1. **Limitations Section.**  Thank you for the suggestion. In the final version, we will include:
>    - (i) a discussion of Assumption 3.3, clarifying its role and possible relaxations;
>    - (ii) limitations of the mixture policy under alternative constraints (e.g., CVaR) and remedies via state augmentation;
>    - (iii) extensions to practical deep RL settings, including efficient oracle implementations.
>
> 2. **Dependence on $\mathcal{C}_\varepsilon(\mathcal{D}, \delta)$.** Intuitively, $\mathcal{C}_\varepsilon(\mathcal{D}, \delta)$ is the minimal number of CMDPs whose $\varepsilon$-neighborhoods cover at least a $(1-\delta)$ fraction of $\mathcal{D}$. It increases as $\varepsilon$ or $\delta$ decreases. Even if $|\Omega|$ is large or infinite, it remains finite (Appendix B) and can be much smaller when $\mathcal{D}$ is concentrated.
>
>    For a 1D Gaussian:
>    $$\mathcal{C}_\varepsilon(\mathcal{D}, \delta) = \mathcal{O}(\varepsilon^{-1}\sigma\sqrt{\ln(1/\delta)}).$$
>    This shows linear dependence on $\varepsilon^{-1}$ and logarithmic dependence on $\delta$. For a uniform distribution over compact $K \subset \mathbb{R}^d$, it is
>    $$ \mathcal{O}\big(\varepsilon^{-d}(1-\delta)|K|\big),$$
>   where $|K|$ denotes the volume of $K$, which typically grows exponentially with the dimension $d$. Thus, complexity grows exponentially in $d$ and linearly in $(1-\delta)$. Since overall sample complexity scales with $\mathcal{C}_\varepsilon(\mathcal{D}, \delta)$, concentrated task distributions significantly reduce test-time learning burden. Moreover, Theorem 5.1 shows that this dependence is tight and cannot be removed.
>
> 3. **Gridworld vs. Safety Gym.**  We will clarify our choice of gridworld and, following your suggestion, have added experiments on Safety Gym. To validate theoretical bounds, we require:
>
>    1. *Exact computation:*  Gridworld enables *exact* optimal policies and exact evaluation of rewards and constraints. This enables precise learning curves and strict verification of our theoretical analysis, free from estimating errors inherent in continuous control settings in Safety-Gym.
>
>    2. *Verifiable assumptions:*  In Gridworld, we can rigorously verify Assumption 3.3. For continuous tasks, verifying feasibility across all tasks is difficult.
>
>    3. *Rigorous adaptation test:*  Constraints are designed so that the optimal policy for each task lies on the boundary of the safe set. This forces the agent to adapt from a feasible policy across all tasks that is initially conservative to a high-reward optimal policy without violating constraints.
>
>    4. *Baseline comparison:*   We compare against Safe Meta-RL[1], LB-SGD [2](model-free RL), and DOPE+[3] (model-based RL), all of which provide safe exploration guarantees. Model-based methods like [3] require solving linear programs over discrete state spaces. Gridworld provides a natural benchmark for the discrete setting, whereas Safety Gym has a continuous state space.
>
>    **Safety Gym experiments.**  Thank you for suggesting continuous control experiments. We evaluate on Hopper locomotion tasks with a 12-dimensional state space and 3-dimensional action space.  The reward is the negative absolute error between velocity and a task-specific goal velocity. Tasks are defined by goal velocities drawn from a truncated distribution over [0,1] with mean 0.5 and variance 0.1. The cost corresponds to control effort.
>
>    We build on implementation in [1] and compare against three baselines over 20 test tasks, each drawn from the task distribution:(a) MAML [4] with constraint penalty, (b) meta CPO [5], (c) CPO [6]. Methods (a) and (b) are constrained meta RL; (c) is online constrained RL.
>
>    Results (see https://osf.io/6435m/overview?view_only=b81ee115ebb34c9297e5791c7dc132de) show that our algorithm is the only method that consistently improves reward while increasing its safety margin. In contrast, MAML with constraints violates safety, CPO exhibits unstable reward learning, and meta-CPO stagnates at a suboptimal, borderline-safe policy due to failure to find feasible updates during testing.
>
>   Due to time constraints, we were not able to tune SafeMeta [1] to ensure safe exploration during testing. We used the publicly available implementation and its suggested hyperparameters, yet we were unable to reproduce its safe exploration performance with such choices. We expect that improved hyperparameter tuning may resolve this issue. We are currently working on this and will include the results in the final manuscript.
>
> [1] Efficient safe meta-reinforcement learning: Provable near-optimality and anytime safety
> [2] A safe exploration approach to CMDPs
> [3] Improved regret bounds for safe RL
> [4] Model-agnostic meta-learning
> [5] Constrained meta-RL with differentiable convex programming
> [6] Constrained policy optimization

---

> > ### Author Rebuttal · Reviewer_65Bn · 2026-04-01
> >
> > Thank you for the responses, but let me choose *(c) partially resolved or unresolved* though this should be selected sparingly.
> >
> > I have mixed feelings.
> > - As I rated Weak Accept at the timing of the initial review, I think this is a nice theoretical safe RL paper. The theoretical results are solid and this paper would be helpful for other researchers.
> > - On the other hand, after reading the rebuttal, I still feel the empirical experiments are not sufficient. Given safe MetaRL (Xu and Zhu, 2025) is successfully applied to Safety-Gym benchmark, I consider that it is better to compare the proposed method with related work in more complicated benchmark.
> > - Though I appreciate the additional experiment on Hopper task, I do not think the this result on a single task supports the effectiveness of the proposed method. Rather, I even feel that this paper would be strengthened by conducing thorough experiments on such a difficult tasks. However, addressing them requires a significant update to the paper, and it is impossible to do that in a short authors-reviewers discussion period.
> >
> > I feel that acceptance or rejection is possible depending on whether the strengths of the theory are highlighted or the weaknesses of the experiments are a concern.

---

> > > ### Author Response · Authors · 2026-04-07
> > >
> > > Thank you for your thoughtful feedback and for highlighting the importance of stronger empirical validation. We are glad that the theoretical contributions were well received; indeed, the main goal of this work is to provide the first end-to-end guarantees, and the experimental section was initially designed to illustrate and validate the key assumptions within the limited space.
> > >
> > > Following your suggestion, we have significantly expanded the empirical evaluation during the rebuttal phase. In particular, we implemented additional verification in Safety Gym and were able to evaluate two scenarios (Hopper and Half-Cheetah). A summary of these results and comparisons is provided below.
> > >
> > > We compare against four baselines: (a) MAML [1] with a constraint penalty, (b) meta-CPO [2], (c) SafeMeta [3], and (d) CPO [4]. Methods (a, b, c) are constrained meta-RL approaches, while (d) is a standard online RL algorithm. As shown in the results (see https://osf.io/6435m/overview?view_only=b81ee115ebb34c9297e5791c7dc132de), all our experiments demonstrate that our method is the only approach that achieves superior optimality while ensuring safe exploration, and we expect this observation to hold across other settings.
> > >
> > > We agree that further empirical validation would be valuable, and we are actively working toward extending these experiments to additional Safety Gym environments and related benchmarks. We would be happy to include these extended results in the appendix or supplementary material in the final version.
> > >
> > > At the same time, given the primary focus of the paper on theoretical guarantees, our intention is for the current set of experiments to serve as an illustrative validation of the framework. A more extensive empirical study would likely require a separate, more application-focused treatment, which we see as a natural direction for follow-up work.
> > >
> > > [1] Finn, Chelsea, Pieter Abbeel, and Sergey Levine. Model-agnostic meta-learning for fast adaptation of deep networks. International conference on machine learning. PMLR, 2017.
> > >
> > > [2] Cho, Minjae, and Chuangchuang Sun. Constrained meta-reinforcement learning for adaptable safety guarantee with differentiable convex programming. Proceedings of the AAAI Conference on Artificial Intelligence. Vol. 38. No. 19. 2024.
> > >
> > > [3] Xu, Siyuan, and Minghui Zhu. Efficient Safe Meta-Reinforcement Learning: Provable Near-Optimality and Anytime Safety. The Thirty-ninth Annual Conference on Neural Information Processing Systems. 2025..
> > >
> > > [4] Achiam, Joshua, David Held, Aviv Tamar, and Pieter Abbeel. Constrained policy optimization. In International conference on machine learning, pp. 22-31. Pmlr, 2017.

---

### Official Review · Reviewer_Q6RJ · 2026-03-13

**Soundness:** 2
**Presentation:** 2
**Significance:** 2
**Originality:** 2
**Overall Recommendation:** 4
**Confidence:** 2

**Summary:**

The paper presents an algorithm for safe meta-reinforcement learning that trains on many constrained tasks and must safely handle a new, unseen task at test time. During training, the method learns a set of near-optimal policies for different types of tasks and also learns a globally safe fallback policy that always satisfies the constraints. When the agent is deployed on a new task, it adaptively mixes a candidate near-optimal policy with the safe fallback policy at every step, ensuring that the constraints are never violated. The authors prove that this mixture remains safe throughout the interaction. The key theoretical result shows that the number of training samples needed depends on a covering number that measures how diverse the task distribution is, rather than the much larger size of the state–action space, meaning learning can be far more efficient when tasks are similar. They also prove a matching lower bound, showing the method is close to optimal in terms of sample complexity.

**Compliance With Llm Reviewing Policy:**

Affirmed.

**Final Justification:**

The paper is theoretically solid but lacks rigorous empirical evaluation on standard safe exploration benchmarks

**Key Questions For Authors:**

1. Since Meta-RL is meant to increase the test-time efficiency of the RL algorithms, results on more siginifcant benchmarks such as Mujoco or Gym environments even with discrete-action space can be valuable justification of the practicality of this approach.

2. Comparision with [1] and [2] should be performed on both the gridworld and the Mujoco/Gym environments.

[1] Khattar et al. A CMDP-WITHIN-ONLINE FRAMEWORK FOR META-SAFE REINFORCEMENT LEARNING ICLR 2023

[2] Xu, S. and Zhu, M. Efficient safe meta-reinforcement learning: Provable near-optimality and anytime safety. NeurIPS 2025

**Limitations:**

Limitations in Questions & Weaknesses

**Strengths And Weaknesses:**

## Strengths:

1. Safe deployment under distribution shift — training in simulation, deploying in the real world with strict constraint satisfaction — is one of the most practically significant problems in safe RL. The paper formalizes this setting rigorously and produces the first algorithm with end-to-end provable guarantees for it.

2. For concentrated task distributions (e.g., sub-Gaussian parameter distributions), $C_ε(D,δ)$ can be exponentially smaller than |S||A|. The example with a d-dimensional Gaussian (reducing to $Õ(ε⁻¹σ√ln δ⁻¹)$ in one dimension) illustrates when meta-training yields practical benefit. The benefit is correctly quantified: when D is broad (e.g., d-dimensional uniform), $C_ε$ grows as $ε⁻d$ and training offers little gain.



## Weaknesses:

1. Assumption 3.3 (Simultaneous Slater's Condition) requires the existence of a "single" policy $π$ that is $ξ-$feasible for "every" CMDP in M_all — not just for tasks drawn from D, but for the entire support of the task family. This assumption is the central pillar of the entire approach: without it, Algorithm 2 has no safe fallback policy π_s to blend with, and safe exploration cannot be guaranteed.

For example for Autonomous driving with task-specific speed limits

  Tasks: different roads with different speed limits (school zones at 15 mph, highways at 65 mph). Constraint: don't exceed the task-specific speed limit.
  $π_s$ must satisfy all speed limits simultaneously — meaning it must always drive at 15 mph (the binding constraint). It can never use information about the current road. Again, $ξ$ is set by the tightest constraint across the entire task family, and the globally safe policy is near-useless for most tasks.

2. The sole experiment is a 7×7 gridworld with a single scalar constraint and a one-dimensional task parameter (slip probability drawn from N(0.3, 0.03)). This setting trivially satisfies Assumption 3.3 — all tasks share the same grid structure, so a globally feasible policy is easy to construct — and falls entirely within the finite CMDP regime where the algorithm applies without approximation. No continuous state/action domain is tested (where the generative model assumption and oracle implementations become non-trivial), no task distribution that stresses Assumption 3.3 is used (e.g., varying obstacle locations), and no sensitivity analysis on ξ or ε is provided.

---

> ### Author Rebuttal · Authors · 2026-03-31
>
> We thank the reviewer for the constructive feedback and appreciate the recognition of the novelty and theoretical contributions of our work. We address the questions below.
>
> 1. **Assumption 3.3.**  We acknowledge that Assumption 3.3 is strong. However, it is necessary, as explained below, and is standard in works providing safe exploration guarantees for constrained meta RL [1,2]. We clarify its role and discuss possible relaxations.
>
>    1. *Necessity.*  Assumption 3.3 requires a feasible policy with a uniform safety margin across tasks (a “Slater policy”). Since the agent operates in the real world without knowing a priori which task it is sampled from, safe exploration requires starting with a policy that is feasible across all tasks.
>
>    2. *Relaxation.*   In Appendix F, we prove that if the objective is relaxed to learning an $\ epsilon$-optimal and $\epsilon$-feasible policy *without* safe exploration guarantees, Assumption 3.3 can be weakened to require only a $\xi$-feasible policy per task, rather than a common policy across all tasks. Under this weaker assumption, convergence follows by adapting the algorithm in [3].
>
>    3. *Finding a Slater policy.*  For finite CMDPs, prior work provides efficient methods to learn a Slater policy [1,2].  For example, [1] proposes a robust policy gradient approach minimizing worst-case constraint violation. If the constraint is shared, robust MDP methods [4,5] can be used:
>       $$ \max_{\pi}\min_{P_i, i\text{ in finite set}} V_c^\pi(P_i),$$
>    which seeks a policy robust to worst-case dynamics. Alternatively, a Slater policy can be obtained via safe learning from demonstrations [6] or by leveraging prior knowledge. For example, as noted by Reviewer Q6RJ, in autonomous driving with task-specific speed limits (e.g., school zones at 15 mph and highways at 65 mph), a low-speed policy satisfies the tightest constraint across all tasks.
>
>    4. *Conservativeness.*  Reviewer Q6RJ raises the concern that Assumption 3.3 may lead to a conservative “safe fallback” policy. In the driving example, a low-speed policy is conservative on highways but ensures feasibility. Our mixture policy mitigates this by allowing the agent to safely increase its speed as it gathers information during testing.
>
> 2. **Baseline comparisons ([1,7]).**  Thank you for the suggestion. We are currently conducting experiments in Safety Gym, comparing against [1,7].  We already compared against[1] in gridworld (Figure 3), labeled *Safe Meta-RL*. All methods ensure safe exploration, but our method achieves about 50% lower test-time reward regret. Our work focuses on safe exploration, whereas [7] learns $\epsilon$-optimal policies *without* safe exploration guarantees, so it was not included in the initial submission.
>
> 3. **Discussion on experiments.**  Thank you for suggesting continuous control experiments. We evaluate on Hopper locomotion tasks with a 12-dimensional state space and 3-dimensional action space.  The reward is the negative absolute error between velocity and a task-specific goal velocity. Tasks are defined by goal velocities drawn from a truncated distribution over [0,1] with mean 0.5 and variance 0.1. The cost corresponds to control effort.
>
>    We build on implementation in [1] and compare against three baselines over 20 test tasks, each drawn from the task distribution:(a) MAML [8] with constraint penalty, (b) meta CPO [9], (c) CPO [10]. Methods (a) and (b) are constrained meta RL; (c) is online constrained RL. We implement oracles as follows: Slater policy using robust policy gradient [1], Optimal policy oracle by CPO, CMDP check by calculating the difference in goal velocities.
>
>    Results (see https://osf.io/6435m/overview?view_only=b81ee115ebb34c9297e5791c7dc132de) show that our algorithm is the only method that consistently improves reward while increasing its safety margin. In contrast, MAML with constraints violates safety, CPO exhibits unstable reward learning, and meta-CPO stagnates at a suboptimal, borderline-safe policy due to failure to find feasible updates during testing.
>
>    Due to time constraints, we were unable to tune SafeMeta[1] to ensure safe exploration. We used the authors' publicly available implementation and their suggested hyperparameters, yet we were unable to reproduce their safe exploration performance with such choices. We expect that improved hyperparameter tuning may resolve this issue and will include the results in the final manuscript.
> [1] Efficient safe meta-reinforcement learning: Provable near-optimality and anytime safety
> [2] SPiDR: A simple approach for zero-shot safety in sim-to-real transfer
> [3] On the power of pre-training for generalization in RL
> [4] Distributionally robust MDPs
> [5] Robust MDPs
> [6] Learning safety constraints from demonstrations
> [7] A CMDP-within-online framework for meta-safe RL
> [8] Model-agnostic meta-learning
> [9] Constrained meta RL with differentiable convex programming
> [10] Constrained policy optimization

---

> > ### Author Rebuttal · Reviewer_Q6RJ · 2026-04-02
> >
> > I am increasing my score but concerns about evaluation and practicality of the method remain. Waiting for the Safety Gym results.

---

> > > ### Author Response · Authors · 2026-04-07
> > >
> > > Thank you for the update and for taking the time to reconsider your evaluation — we appreciate it.
> > >
> > > In response to your concerns about evaluation and practicality, we have added new results on Safety Gym during the rebuttal phase. In particular, we evaluated our method on two scenarios (Hopper and Half-Cheetah), which provide a more realistic testbed for safety-constrained learning. As shown in the results (summarized here: https://osf.io/6435m/overview?view_only=b81ee115ebb34c9297e5791c7dc132de), all our experiments demonstrate that our method is the only approach that achieves superior optimality while ensuring safe exploration compared to other baselines (a) MAML with a constraint penalty [1], (b) meta-CPO [2], (c) SafeMeta [3], and (d) CPO [4], supporting its practical applicability. We expect this observation to hold across other settings.
> > >
> > > We agree that broader empirical validation would further strengthen the paper, and we are continuing to extend these experiments to additional Safety Gym environments. We would be happy to include these additional results in the appendix or supplementary material in the final version.
> > >
> > >
> > > [1] Finn, Chelsea, Pieter Abbeel, and Sergey Levine. Model-agnostic meta-learning for fast adaptation of deep networks. International conference on machine learning. PMLR, 2017.
> > >
> > > [2] Cho, Minjae, and Chuangchuang Sun. Constrained meta-reinforcement learning for adaptable safety guarantee with differentiable convex programming. Proceedings of the AAAI Conference on Artificial Intelligence. Vol. 38. No. 19. 2024.
> > >
> > > [3] Xu, Siyuan, and Minghui Zhu. Efficient Safe Meta-Reinforcement Learning: Provable Near-Optimality and Anytime Safety. The Thirty-ninth Annual Conference on Neural Information Processing Systems. 2025..
> > >
> > > [4] Achiam, Joshua, David Held, Aviv Tamar, and Pieter Abbeel. Constrained policy optimization. In International conference on machine learning, pp. 22-31. Pmlr, 2017.

---

### Official Review · Reviewer_4oNp · 2026-03-13

**Soundness:** 4
**Presentation:** 2
**Significance:** 3
**Originality:** 3
**Overall Recommendation:** 5
**Confidence:** 4

**Summary:**

This paper addresses the fundamental tension between rapid down-stream adaptation and the satisfaction of safety constraints in constrained meta-reinforcement learning (meta-CRL). It is proposing an algorithmic framework for meta-CRL that comes with both sample efficiency guarantees (in terms of training-time environment interactions) and constraint satisfaction guarantees.

The framework follows the standard meta-RL structure of a training phase and a test phase. During training, the algorithm uses three oracles: (i) a compression oracle that identifies a reduced set of representative CMDPs from the training distribution, (ii) a learning oracle that computes near-optimal policies and value functions for each representative CMDP, and (iii) a "Slater oracle" that finds a policy satisfying all constraints simultaneously with a margin across the representative CMDPs (I'll refer to this policy as the "Slater policy"). Under stated assumptions on oracle quality, the authors establish optimality and feasibility bounds for the Slater policy with respect to the original CMDPs, as well as a sample complexity bound on the number of training CMDP samples required. At test time, the algorithm uses a mixture policy that linearly interpolates between the Slater policy and the highest-reward policy from the training phase.

While some of the assumptions appear restrictive, the framework is well-motivated and well-presented, and is likely to contribute to a better theoretical understanding of the meta-CRL problem.

**Compliance With Llm Reviewing Policy:**

Affirmed.

**Final Justification:**

The authors have addressed my concerns. This is a solid theory paper and I recommend acceptance.

**Key Questions For Authors:**

Q1. Assumption 3.3 requires the existence of a single policy satisfying all representative CMDPs' constraints simultaneously. Can this assumption be relaxed, for example, to the "relaxed xi-feasibility" condition mentioned earlier in the paper? Under what practical conditions does Assumption 3.3 actually hold? It would strengthen the paper to discuss conditions under which the respective oracle can be instantiated.

Q2. The test-time mixture policy linearly interpolates between the Slater policy and a high-reward policy. This is intuitive and analytically tractable, but what are the limitations of this approach? As far as I understand, a linear mixture of two feasible policies is not guaranteed to remain feasible, and the reward achieved by a strictly feasible mixture may be far from the constrained optimum.

Q3. The CMDP distance d(Mi, Mj) does not appear to be reward-scale invariant. Could this cause the compression step to behave poorly when reward scales differ across tasks? Would an alternative distance be more appropriate in some situations?

**Limitations:**

Limitations of the proposed approach in terms of the necessary assumptions for the guarantees to hold and in terms of practical applicability could be discussed more clearly.

**Strengths And Weaknesses:**

- Soundness:
  - Strengths: The theoretical contributions are clearly stated and are technically sound. The sample complexity and feasibility bounds are derived under explicit oracle assumptions, which makes the guarantees transparent and the proof structure easy to follow.
  - Weaknesses / concerns: Assumption 3.3 (the Simultaneous Slater Condition) seems restrictive. It requires the existence of a single policy that satisfies all constraints across all representative CMDPs simultaneously. Even when such a policy exists, it is unclear how efficiently it can be found.
- Presentation:
  - Strengths: The theoretical parts of the paper are well-organized and the high-level framework is clearly motivated. The separation into training and test phases is intuitive, and the role of each oracle is explained clearly. Notation is consistent throughout. The relevant work is adequately discussed.
  - Weaknesses / clarity issues: The presentation is a bit dense, and it would benefit from more discussion and verbal exposition throughout. A discussion section before the conclusions might be useful to discuss advantages and disadvantages of the framework, and limitations of the theoretical analysis as well as practical applicability.
- Significance:
  - Strengths: Meta-CRL is a timely and practically important problem. To the best of my knowledge, the paper provides one of the first formal frameworks with joint sample complexity and constraint satisfaction guarantees in this setting, which is a meaningful contribution to the literature.
  - Weaknesses / limitations: The framework is mostly theoretical, which limits its immediate applicability. The paper would benefit from a more detailed discussion of limitations in this regard and how they could be addressed in future work.
- Originality:
  - Strengths: The combination of meta-learning with constrained RL guarantees, including explicit sample complexity bounds, appears to be a novel contribution. The particular combination of the pretrained "Slater policy" and the test-time mixture policy is an interesting and original design choice.
  - Weaknesses / related-work overlap: It would be helpful to discuss how the feasibility guarantees compare to existing results in standard (non-meta) constrained RL, and what the additional difficulty of the meta-learning component is.

---

> ### Author Rebuttal · Authors · 2026-03-31
>
> We thank the reviewer for the detailed and insightful assessment of our paper. Below, we address the raised concerns and questions.
>
> 1. **Assumption 3.3.**  Please see response 1 to Reviewer Q6RJ.
>
> 2. **Mixture policy.**  We thank the reviewer for raising this important point. Below, we explain how the proposed mixture strategy ensures safe exploration and convergence to optimality, and discuss its limitations. We will incorporate these discussions in the revised manuscript.
>
>    1. *Feasibility.*  A policy $\pi$ is feasible if its expected constraint value satisfies $V_c^\pi \ge 0$. As defined in Definition 2.3, a mixture policy samples a base policy at the beginning of each episode and executes it for the entire trajectory. As a result, both the expected reward and constraint value are linear combinations of those of the base policies. Therefore, any mixture of feasible policies remains feasible.
>
>    2. *Optimality.* The mixing weight $\alpha$ is adaptive and converges exponentially fast to $1 - \mathcal{O}(\varepsilon)$ (Eq. (6)), so the mixture rapidly concentrates on the near-optimal policy.
>
>    3. *Limitations.*   We thank the reviewer for pointing out the missing discussion on limitations. Our safe exploration analysis for the mixture policy relies on the fact that its value is a linear combination of those of the base policies, together with the existence of a Slater policy that provides a strictly positive safety margin. The analysis may break down in the following cases:
>
>       1. *No safety margin.*  If the Slater policy does not provide a strictly positive safety margin (e.g., under equality constraints $V_c^\pi = 0$ or when Assumption 3.3 does not hold), such a policy may not exist. For example, in autonomous driving, if one task requires driving on the left and another on the right, no single Slater policy can satisfy both constraints.
>
>       2. *Nonlinear constraint structure.*  We consider constraints of the form ${E}[\sum_t \gamma^t c(s_t,a_t)]$, consistent with standard CMDP formulations [2,3]. Under this setting, the value of a mixture policy is a linear combination of those of the base policies. For other constraints such as CVaR, this linearity may not hold and our analysis does not directly apply. However, CVaR constraints can be reformulated as expected cumulative constraints via state augmentation[1], in which case our results extend.
>
> 3. **CMDP distance $d(M_i, M_j)$.**  We agree that the dependence of the CMDP distance on the reward scale is important, as large variations in reward magnitude (e.g., up to $R$) can loosen the bounds by a factor of $R$. In our setting, we assume rewards are bounded in $[0,1]$, which controls this effect. This assumption is standard in RL and our results can be extended to a bounded reward setting. For practical settings with heterogeneous reward scales across tasks, standard reward normalization can be applied when constructing each test task since optimal policy under the original reward remains optimal under rescaling. We will clarify this point and include a discussion in the final version.
>
> 4. **Discussion and Limitations section.**  Thank you for pointing out the dense presentation. We will add more discussion and exposition, including
>    - (i) a discussion of Assumption~3.3, clarifying its role and possible relaxations,
>    - (ii) limitations of the mixture policy,
>    - (iii) extensions to practical deep RL settings, including a discussion of efficient implementations of the oracles used in our approach.
>
> 5. **Comparison to constrained RL.**  Meta constrained RL leverages experience across a task distribution to enable faster adaptation on new tasks while satisfying constraints. In contrast, standard constrained RL learns from scratch on each task. Most constrained RL works target $\varepsilon$-optimal and $\varepsilon$-feasible policies *without* safe exploration guarantees [2,3]. Our work explicitly focuses on safe exploration, so we compare only with such methods. As shown in Table 1 and Remark 4.4, existing methods scale with $|S||A|$ (model-based) or poorly with $\varepsilon$ (model-free). In contrast, our approach depends on the covering number $C_\varepsilon$, which can be much smaller for concentrated task distributions, yielding significant sample efficiency gains.
>
> 6. **Challenge in meta constrained RL.**  The key challenge is to leverage training experience to adapt efficiently to unseen CMDPs during testing while ensuring safe exploration and minimal interactions.
>
>    To address this, we learn:
>    - (i) a set of policies approximating optimal policies across tasks, and
>    - (ii) a single Slater policy.
>
>    At test time, we use an adaptive mixture scheme to refine these policies, enabling efficient adaptation with safe exploration guarantees.
>
> [1] Provably efficient CVaR RL in low-rank MDPs
> [2] Natural policy gradient primal-dual method for constrained MDPs
> [3] A near-optimal primal-dual method for off-policy learning in CMDP

---

> > ### Author Rebuttal · Reviewer_4oNp · 2026-04-02
> >
> > The authors have addressed all three of my questions. The clarification that mixture policy feasibility follows from the episode-level sampling scheme (making constraint values linear in the base policies) resolves my concern. I had implicitly assumed a step-level mixture, but the episode-level mixture was in fact clearly stated in Definition 2.3. The reward-scale invariance concern is also resolved: I missed that rewards are bounded in [0,1], which I should have caught from the paper. I maintain my score.

---

> > > ### Author Response · Authors · 2026-04-07
> > >
> > > Thank you for your answer. We are happy to hear that your concerns have been resolved.

---

### Decision · Program_Chairs · 2026-04-30

**Decision:**

Accept (regular)

**Comment:**

Reviewers reach a consensus in recommending acceptance for this paper. Reviewers agree that this paper has the following strengths:

1. This paper studies an important problem in safe RL, i.e., training in simulation, and deploying in the real world with strict constraint satisfaction. The paper formalizes this setting rigorously and produces the first algorithm with end-to-end provable guarantees for it.
2. While theorems look somewhat incremental from those in typical safe RL literature, several results such as Theorems 4.2 and 5.1 are useful and new.

Reviewers also note several limitations:

1. Given safe MetaRL (Xu and Zhu, 2025) is successfully applied to Safety-Gym benchmark, it is better to compare the proposed method with related work in more complicated benchmark. This paper would be strengthened by conducing thorough experiments.

Overall, this paper makes solid contributions to constrained meta RL and merits acceptance.